# FLAC: Maximum Entropy RL via Kinetic Energy Regularized Bridge Matching

**Lei Lv** [1 2 3]  **Yunfei Li** [2]  **Yu Luo** [3]  **Fuchun Sun** [3]  **Xiao Ma** [2]

## Abstract

Iterative generative policies, such as diffusion models and flow matching, offer superior expressivity for continuous control but complicate Maximum Entropy Reinforcement Learning because their action log-densities are not directly accessible. To address this, we propose **Field Least-Energy Actor-Critic (FLAC)**, a likelihood-free framework that regulates policy stochasticity by penalizing the kinetic energy of the velocity field. Our key insight is to formulate policy optimization as a Generalized Schrödinger Bridge (GSB) problem relative to a high-entropy reference process (e.g., uniform). Under this view, the maximum-entropy principle emerges naturally as staying close to a high-entropy reference while optimizing return, without requiring explicit action densities. In this framework, kinetic energy serves as a physically grounded proxy for divergence from the reference: minimizing path-space energy bounds the deviation of the induced terminal action distribution. Building on this view, we derive an energy-regularized policy iteration scheme and a practical off-policy algorithm that automatically tunes the kinetic energy via a Lagrangian dual mechanism. Empirically, FLAC achieves superior or comparable performance on high-dimensional benchmarks relative to strong baselines, while avoiding explicit density estimation.

## 1. Introduction

Iterative generative policies, including flow matching and diffusion models (Dhariwal & Nichol, 2021; Lipman et al., 2022; Ho et al., 2020), have recently emerged as a powerful paradigm in reinforcement learning (Wang et al., 2022; Park et al., 2025). Unlike conventional Gaussian actors (Haarnoja et al., 2018) that output actions directly, these implicit policies define the policy through a sequential generation procedure that transport a simple base noise distribution to complex, state-conditioned action distributions. This expressiveness allows for modeling rich, multi-modal behaviors (Chi et al., 2023), enabling these policies to achieve superior performance in high-dimensional control tasks and data-driven settings where simple unimodal distributions fall short.

However, coupling these iterative generative policies with Maximum-Entropy RL (Ziebart, 2010; Haarnoja et al., 2018) is nontrivial. In RL, a Maximum-Entropy objective is often essential for preventing premature collapse and for sustaining exploration by explicitly encouraging stochasticity. Yet Maximum-Entropy methods typically rely on the policy log-density $\log \pi(a \mid s)$ to quantify and regulate this stochasticity. For iterative generators, $\log \pi(a \mid s)$ is not directly accessible and is often difficult to compute since the action distribution is only defined implicitly through a multi-step generation procedure. Consequently, existing approaches resort to additional estimation machinery (Celik et al., 2025), such as training auxiliary networks (Zhang et al., 2025) or regularizing tractable distributional proxies (Wang et al., 2024). While effective in some cases, these strategies introduce extra complexity and computation, and often lead to suboptimal exploration.

To address this, we propose a fundamental shift in perspective: instead of explicitly estimating and tuning terminal entropies, we cast entropy-regularized *policy optimization* as a Generalized Schrödinger Bridge (GSB) problem (Liu et al., 2024). The Schrödinger Bridge Problem (SBP) (Pavon et al., 2021; Shi et al., 2023) studies entropy-regularized transport by finding a trajectory distribution that stays close to a reference stochastic process while inducing desired terminal behavior. In this framework, the Maximum Entropy principle is no longer an external heuristic; rather, it follows from a structured trade-off between terminal utility and closeness to a high-entropy reference on path space. In particular, our derivation characterizes the induced terminal action distribution as a reweighting of the reference terminal marginal; when this reference marginal is set to be approximately uniform over the bounded action domain, the characterization

[1]Shanghai Research Institute for Intelligent Autonomous Systems, Shanghai, China [2]ByteDance Seed, Beijing, China [3]Department of Computer Science and Technology, Tsinghua University, Beijing, China. Correspondence to: Fuchun Sun <fcsun@tsinghua.edu.cn>, Xiao Ma <xiao.ma@bytedance.com>.

aligns with the standard maximum-entropy principle. Crucially, we theoretically show that controlling deviation from the reference on the path space also controls the induced terminal action distribution. Moreover, for velocity-field-driven iterative generators, we show that this path-space deviation can be controlled via the kinetic energy of the flow (Liu et al., 2024) (i.e., the expected path integral of the squared velocity/drift magnitude along the generation trajectory), which directly motivates a least-kinetic regularizer.

Motivated by this perspective, we propose **Field Least-Energy Actor-Critic (FLAC)**, a novel framework that instantiates this least-kinetic GSB regularization in RL. The actor is optimized to maximize Q-values while simultaneously minimizing this kinetic energy, effectively balancing reward maximization with the preservation of generation stochasticity. Furthermore, we introduce an automatic tuning mechanism for the energy penalty, ensuring the policy adapts its exploration level dynamically during training.

We evaluate FLAC on a suite of challenging continuous control benchmarks, including DMControl (Tassa et al., 2018) and HumanoidBench (Sferrazza et al., 2024). Our results demonstrate that FLAC achieves competitive or superior performance compared to state-of-the-art baselines.

## 2. Preliminaries

### 2.1. Entropy-Regularized RL

We consider a Markov Decision Process (MDP) (Bellman, 1957) defined by the tuple $\mathcal{M} = (\mathcal{S}, \mathcal{A}, p, r, \gamma)$, with continuous state space $\mathcal{S} \in \mathbb{R}^{d_s}$ and action space $\mathcal{A} \in \mathbb{R}^{d_a}$. The transition dynamics are $p(s' \mid s, a)$, the reward function is $r(s, a)$, and $\gamma \in [0, 1)$ is the discount factor. The goal is to learn a policy $\pi(a \mid s)$ that maximizes the expected return (Sutton et al., 1998).

In continuous control, to prevent premature convergence and encourage exploration, the objective is often augmented with an entropy term (Maximum Entropy RL):

$$J_{\text{MaxEnt}}(\pi) = \mathbb{E}_\pi \left[ \sum_{t=0}^{\infty} \gamma^t (r(s_t, a_t) + \alpha \mathcal{H}(\pi(\cdot \mid s_t))) \right], \quad (1)$$

where $\mathcal{H}(\pi) = -\mathbb{E}_{a \sim \pi}[\log \pi(a \mid s)]$, and maximizing $\mathcal{H}(\pi)$ is equivalent to minimizing $D_{\text{KL}}(\pi(\cdot \mid s) \,\|\, \text{Unif}(\mathcal{A}))$. Notably, MaxEnt RL yields a Boltzmann optimal policy of the form $\pi^*(a \mid s) \propto \exp(Q(s, a)/\alpha)$, mirroring the exponential-tilting closed-form structure that will reappear in our GSB formulation.

### 2.2. Iterative Generative Policies

Unlike explicit policies (e.g., Gaussians) that directly output action samples, iterative generative policies define the distribution $\pi(a \mid s)$ implicitly through a transport process. Let

$\tau \in [0, 1]$ denote the continuous generation time. The action generation is modeled as the solution to a state-conditioned Stochastic Differential Equation (SDE) (Song et al., 2020; Liu et al., 2025):

$$dX_\tau = u(s, \tau, X_\tau)d\tau + \sigma dW_\tau, \quad X_0 \sim \mu_0, \quad (2)$$

where $X_\tau \in \mathbb{R}^{d_a}$ is the latent state, $X_0$ is sampled from a simple prior $\mu_0$ (typically $\mathcal{N}(0, I)$ or uniform distribution), and $a := X_1$ is the realized action. The drift term $u_\theta : \mathcal{S} \times [0, 1] \times \mathbb{R}^{d_a} \to \mathbb{R}^{d_a}$ is a learnable vector field (velocity field), and $W_\tau$ is a standard Wiener process.

A key property of Eq. (2) is that the marginal density of the terminal state, $\pi(X_1 \mid s)$ is not directly accessible. Evaluating $\log \pi(a \mid s)$ requires solving the instantaneous change of variables formula or marginalizing over all possible paths, which is computationally expensive and numerically unstable during online training. This necessitates a likelihood-free approach to stochasticity regulation.

### 2.3. The Schrödinger Bridge Problem

The Schrödinger Bridge (SB) problem addresses the question of finding the most likely stochastic evolution between two probability distributions given a reference process. Formally, let $\Omega = C([0, 1], \mathbb{R}^d)$ be the path space, and let $X_\tau : \Omega \to \mathbb{R}^d$ be the canonical coordinate process defined by $X_\tau(\omega) = \omega(\tau)$, where $\omega \in \Omega$. We denote the marginal distribution at time $\tau$ as

$$\mathbb{P}_\tau := (X_\tau)_\# \mathbb{P}.$$

Given a reference $\mathbb{P}^{\text{ref}}$ (typically the uncontrolled Brownian motion) (Léonard, 2013) and two marginals $\mu_0, \mu_1$, the SB problem seeks a measure $\mathbb{P}^*$ that minimizes a divergence metric $\mathcal{D}$ with respect to the reference, subject to matching the marginals:

$$\min_{\mathbb{P}} \ \mathcal{D}(\mathbb{P} \| \mathbb{P}^{\text{ref}}) \quad \text{s.t.} \quad \mathbb{P}_0 = \mu_0, \ \mathbb{P}_1 = \mu_1. \quad (3)$$

Specifically, for SDEs, $\mathcal{D}$ is the KL divergence; for ODEs, it connects to the Wasserstein-2 distance (Tamogashev & Malkin, 2025). This formulation is often referred to as a "Data-to-Data" bridge, commonly used in generative modeling to connect noise and data. Recent works (Liu et al., 2024) have extended this to the Generalized Schrödinger Bridge (GSB), where the hard terminal constraint $\mathbb{P}_1 = \mu_1$ is relaxed into a soft potential or functional constraint. This generalization is crucial for our formulation in Section 3, where the target is defined by rewards rather than samples.

### 2.4. Kinetic Energy and Path Divergence

To regulate the policy without access to terminal log-densities, we lift the perspective from the *action space* to

the *path space*. We define the Kinetic Energy of the generation process as the expected physical work done by the drift field:

$$\mathcal{E}(s) := \mathbb{E}\left[\int_0^1 \frac{1}{2}\|u_\theta(s, \tau, X_\tau)\|^2 d\tau\right]. \qquad (4)$$

This quantity serves as a unified proxy for the divergence from the reference measure $\mathbb{P}^{\mathrm{ref}}$ (the base noise process) across both stochastic and deterministic regimes.

**Stochastic Regime ($\sigma > 0$).** The path divergence is proportional to the energy (Tzen & Raginsky, 2019). As derived in Appendix A.1:

$$D_{\mathrm{KL}}(\mathbb{P}^\theta\|\mathbb{P}^{\mathrm{ref}}) = \frac{1}{\sigma^2}\mathcal{E}(s). \qquad (5)$$

Here, $\mathbb{P}^\theta$ and $\mathbb{P}^{\mathrm{ref}}$ denote the policy and reference path measures (both initialized with $X_0 \sim \mu_0$), and their terminal marginals at $\tau = 1$ are $\pi_\theta(\cdot \mid s)$ and $\mu_1^{\mathrm{ref}}$. Crucially, we establish that the divergence between path measures strictly upper-bounds the divergence between $\pi(\cdot|s)$ and the reference terminal marginal $\mu_1^{\mathrm{ref}}$:

$$D_{\mathrm{KL}}(\pi_\theta\|\mu_1^{\mathrm{ref}}) \leq D_{\mathrm{KL}}(\mathbb{P}^\theta\|\mathbb{P}^{\mathrm{ref}}) = \frac{1}{\sigma^2}\mathcal{E}(s). \qquad (6)$$

We provide the proof of this inequality in Appendix A.3. This theoretical result is fundamental to our framework, as it guarantees that minimizing the kinetic energy is a sufficient condition to enforce the constraint on the terminal action distribution.

**Deterministic Regime ($\sigma \to 0$).** In the ODE case, the kinetic energy relates to the Optimal Transport cost (Mikami, 2004; Benamou & Brenier, 2000). As detailed in Appendix A.2:

$$\mathcal{W}_2^2(\mu_0, \pi_\theta) \leq 2\mathcal{E}(s). \qquad (7)$$

In the deterministic (ODE) case, the reference dynamics keeps $X_\tau = X_0$, hence $\mu_1^{\mathrm{ref}} = \mu_0$. Note, while ODE flow is deterministic, the randomness comes from $X_0$. Minimizing kinetic energy acts as a geometric regularizer that strictly bounds the deviation (in Wasserstein-2 distance) from this prior. When $\mu_0$ is uniform over a bounded action domain, this follows a similar principle to maximum-entropy RL, namely discouraging overly concentrated action distributions and encouraging broadly supported, stochastic policies over the action domain, although it does not provide a strict entropy guarantee in the deterministic limit as we discussed in Appendix A.2.

Thus, minimizing kinetic energy consistently enforces closeness to the prior, interpreted as entropic proximity (in SDEs) or geometric proximity (in ODEs). Hence, minimizing this path energy is sufficient to bound the divergence of the terminal action distribution.

## 3. Reinforcement Learning as a Generalized Schrödinger Bridge Problem

In this section, we formally derive FLAC. We begin by reframing the policy optimization problem not merely as maximizing returns, but as a *Generalized Schrödinger Bridge (GSB)* problem. This perspective unifies the generative dynamics and the exploration objective into a single, coherent physical transport formulation.

### 3.1. The Generalized Schrödinger Bridge Formulation

Standard RL treats the policy as a conditional distribution. Here, we view it as a controlled stochastic process. Following the formulation in Liu et al. (2024), we define our goal as finding a path measure $\mathbb{P}$ on the space of trajectories that minimizes a composite objective: a divergence cost relative to a high-entropy reference process, and a terminal potential cost reflecting the task reward.

Let $\mathbb{P}^{\mathrm{ref}}$ denote a fixed reference path measure (e.g., Brownian motion) starting from a high-entropy prior $\mu_0$ (instantiated as a uniform distribution). We formulate the One-Ended Generalized Schrödinger Bridge problem as

$$\min_{\mathbb{P}} \quad \mathcal{J}_{\mathrm{GSB}}(\mathbb{P}) := \alpha \underbrace{\mathcal{D}(\mathbb{P}\|\mathbb{P}^{\mathrm{ref}})}_{\text{Divergence Cost}} + \underbrace{\mathbb{E}_{X_1 \sim \mathbb{P}}[\mathcal{G}(X_1)]}_{\text{Terminal Potential}} \qquad (8)$$
$$\text{s.t.} \quad \mathbb{P}_0 = \mu_0.$$

This optimization is subject to specific boundary conditions that distinguish it from classical transport problems. First, the process is anchored at a fixed start, constrained to initialize from the reference prior $\mu_0$. Second, unlike the standard Schrödinger Bridge which imposes a hard constraint on the terminal marginal (i.e., forcing $X_1$ to match a data distribution), our formulation is one-ended (or "free-end"): the terminal distribution $\mathbb{P}_1$ is free to evolve, regularized only by the soft potential $\mathcal{G}(X_1)$.

We analyze the theoretical properties of this formulation. The optimization problem in Eq. (8) admits a closed-form solution for the terminal marginal distribution.

**Proposition 3.1** (Optimal GSB Solution)**.** *The optimal path measure $\mathbb{P}^*$ that minimizes Eq. (8) induces a terminal marginal distribution $p^*(X_1)$ of the form:*

$$p^*(X_1) \propto \mu_1^{\mathrm{ref}}(X_1) \cdot \exp\left(-\frac{\mathcal{G}(X_1)}{\alpha}\right), \qquad (9)$$

*where $\mu_1^{\mathrm{ref}}(X_1)$ is the marginal distribution of the reference process at $\tau = 1$.*

*Proof.* See Appendix A.4. $\qquad\square$

Proposition 3.1 reveals an exponential-tilting closed form for the optimal terminal marginal. When $\mu_1^{\mathrm{ref}}$ is approxi-

mately uniform over a bounded action domain, the solution reduces to $p^*(X_1) \propto \exp(-\mathcal{G}(X_1)/\alpha)$.

To connect this general form to RL, we introduce a *state-conditioned* terminal potential $\mathcal{G}_s(X_1)$, so that the induced terminal marginal defines a policy $\pi(\cdot \mid s)$ over actions $a := X_1$. In particular, we will instantiate $\mathcal{G}_s(\cdot)$ using a critic-like, value-informed potential (lower potential for higher-value actions), yielding a Boltzmann-style policy family like SAC (Haarnoja et al., 2018):

$$\pi(a \mid s) \propto \mu_1^{\text{ref}} \cdot \exp\left(-\frac{\mathcal{G}_s(a)}{\alpha}\right).$$

### 3.2. Energy-Regularized Policy Optimization

While Proposition 3.1 characterizes the optimal equilibrium, directly sampling from the unnormalized Boltzmann distribution is intractable in high-dimensional continuous spaces. Therefore, we solve the variational problem (Eq. 8) directly by parameterizing the generation process and instantiating the abstract GSB components into a tractable RL objective.

**Deriving the FLAC Objective.** First, leveraging the connection established in Section 2.4, we substitute the abstract divergence term with the expected kinetic energy of the velocity field:

$$\mathcal{D}(\mathbb{P}^\theta \| \mathbb{P}^{\text{ref}}) \propto \mathbb{E}\left[\int_0^1 \frac{1}{2} \|u_\theta\|^2 d\tau\right].$$

Second, to align with the actor-critic framework, we instantiate the terminal potential as the negative *(expected) discounted return* after taking action $a := X_1$ at state $s$:

$$\mathcal{G}_s(X_1) := -R(s, X_1) = -\mathbb{E}\left[\sum_{t=0}^T \gamma^t r(s_t, a_t)\right].$$

Substituting these terms into Eq. 8, we obtain the training objective for our proposed method, Field Least-Energy Actor-Critic (FLAC):

$$\min_\theta J_{\text{FLAC}}(\theta) = \mathbb{E}_{\mathbb{P}^\theta}\left[\underbrace{\alpha \int_0^1 \frac{1}{2} \|u_\theta(s, \tau, X_\tau)\|^2 d\tau}_{\text{Minimize Kinetic}} \right.$$
$$\left. \underbrace{-R(s, X_1)}_{\text{Maximize Return}}\right], \quad \text{s.t. } X_0 \sim \mu_0. \quad (10)$$

Here, the expectation is taken over the trajectory generated by the current policy. The term "Least-Kinetic" reflects the physical intuition of our approach: the kinetic energy term acts as a dynamic regularizer. Since the reference process (Brownian motion) has zero drift (zero kinetic energy),

minimizing energy compels the policy to adhere to the intrinsic stochasticity of the reference, exerting effort *only* when necessary to steer towards high-value regions.

To demonstrate the efficacy of this regularization, we visualize the evolution of the learned vector fields on a 2D multi-goal toy environment (Figure 1). In the Naive Flow case (Top), the policy maxmizes reward without regularization. As observed during the learning progress, it learns an aggressive, high-velocity field (depicted by long red arrows) that rapidly concentrates probability mass. This high kinetic energy completely overpowers the noise, causing the action distribution to suffer from severe mode collapse, capturing only a single goal. In contrast, FLAC (Bottom) penalizes the kinetic energy. The resulting field exerts minimal control effort, indicated by the subtle, low-magnitude field vectors. In the end of training, FLAC successfully maintains sufficient stochasticity to cover all 8 optimal modes, validating our hypothesis that limiting kinetic energy prevents the premature elimination of diverse solution paths.

## 4. Field Least-Energy Actor-Critic

Building on the GSB formulation, we propose **Field Least-Energy Actor-Critic (FLAC)**, which optimizes a velocity field to transport the prior noise to high-reward regions with minimal kinetic energy. This section details the practical algorithm, deriving a rigorous energy-regularized policy iteration scheme and its implementation with automatic energy tuning.

### 4.1. Energy-Regularized Policy Iteration

We incorporate the kinetic energy penalty directly into the Bellman operator. This allows us to extend standard Policy Iteration guarantees to our setting. Analogous to SAC, which derives a soft Bellman backup with an entropy regularizer, we derive an energy-regularized Bellman operator by incorporating the kinetic-energy cost of the action-generation process into the backup.

**Policy Evaluation.** For a fixed policy $\pi$, we define the energy-regularized Bellman evaluation operator $\mathcal{T}^\pi$ acting on $Q : \mathcal{S} \times \mathcal{A} \to \mathbb{R}$ as

$$(\mathcal{T}^\pi Q)(s, a) := r(s, a) + \gamma \mathbb{E}\big[Q(s', a') - \alpha \mathcal{E}_\pi(s')\big], \quad (11)$$

where $\mathcal{E}_\pi(s')$ denotes the expected kinetic energy required to sample $a' \sim \pi(\cdot \mid s')$.

**Proposition 4.1** (Convergence of Policy Evaluation). *Assume rewards are bounded and the energy term is finite. The operator $\mathcal{T}^\pi$ is a $\gamma$-contraction in the $L^\infty$ norm. Consequently, the iterative update $Q_{k+1} = \mathcal{T}^\pi Q_k$ converges to the unique regularized value function $Q^\pi$.*

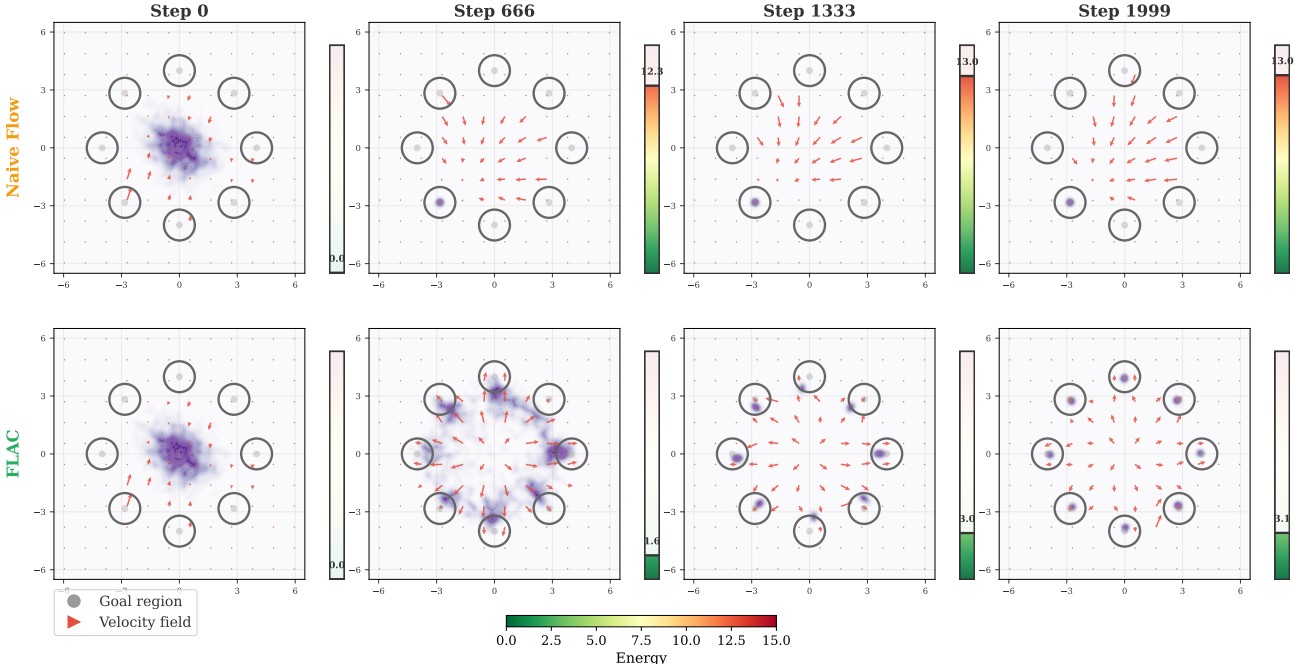

*Figure 1.* Kinetic Energy Regularization Encourage Exploration. Toy example on a 2D multi-goal landscape. (Top) Unconstrained: The high-velocity field overpowers the intrinsic noise, forcing the policy to collapse into a single deterministic mode. (Bottom) FLAC: By penalizing kinetic energy, the policy is constrained to preserve stochasticity. This low-energy field successfully recovers the full multimodal distribution.

*(Proof in Appendix A.5)*

**Policy Improvement.** Given the value function $Q^\pi$, we update the policy to maximize the regularized objective. This corresponds to finding a policy that maximizes the expected Q-value while minimizing its generation energy:

$$\pi \leftarrow \arg\max_\pi \mathbb{E}_{s\sim\mathcal{D}} \left[ \mathbb{E}_{a\sim\pi(\cdot|s)}[Q^\pi(s,a)] - \alpha\mathcal{E}_\pi(s) \right]. \tag{12}$$

**Proposition 4.2** (Monotonic Improvement). *The update rule guarantees monotonic improvement of the generalized objective, i.e., $J_{\mathrm{GSB}}(\pi_{new}) \geq J_{\mathrm{GSB}}(\pi)$. This drives the policy towards the optimal transport plan that balances reward maximization and entropic exploration.*

*(Proof in Appendix A.6)*

### 4.2. Practical Implementation

We instantiate the above framework into a practical off-policy actor-critic algorithm. We parameterize the vector field $u_\theta(s, \tau, X_\tau)$ (Actor) and the state-action value function $Q_\psi(s, a)$ (Critic).

**Critic Update.** The critic is trained to minimize the Bellman residual derived from Eq. (11). To estimate the target value, we sample the next action $a'$ from the current policy at state $s'$ using a numerical solver, and simultaneously

compute its discretized kinetic energy $\widehat{\mathcal{E}}_\theta(s')$. The target value $y$ is constructed as:

$$y = r + \gamma \left( \min_{i=1,2} Q_{\bar\psi_i}(s', a') - \alpha\widehat{\mathcal{E}}_\theta(s') \right), \tag{13}$$

where $Q_{\bar\psi_i}$ are the target critic networks. The critic parameters $\psi$ are updated by minimizing the Bellman Error.

**Actor Update.** The actor updates $\theta$ to maximize the improvement objective. Since the action $a_\theta$ is generated via a differentiable solver, we can backpropagate gradients from the critic through the entire generation trajectory (pathwise derivative). The actor loss is:

$$J_\pi(\theta) = \mathbb{E}_{s\sim\mathcal{B}} \left[ \alpha\widehat{\mathcal{E}}_\theta(s) - Q_\psi(s,a) \right], \tag{14}$$

where $a \sim \pi_\theta(\cdot|s)$. Minimizing this loss encourages the velocity field to find trajectories that lead to high-value actions while maintaining low kinetic energy.

### 4.3. Automatic Energy Tuning

Selecting a fixed regularization coefficient $\alpha$ is challenging, as the magnitude of kinetic energy varies significantly across different tasks and training stages. A fixed $\alpha$ may lead to over-exploration or premature convergence to deterministic behavior.

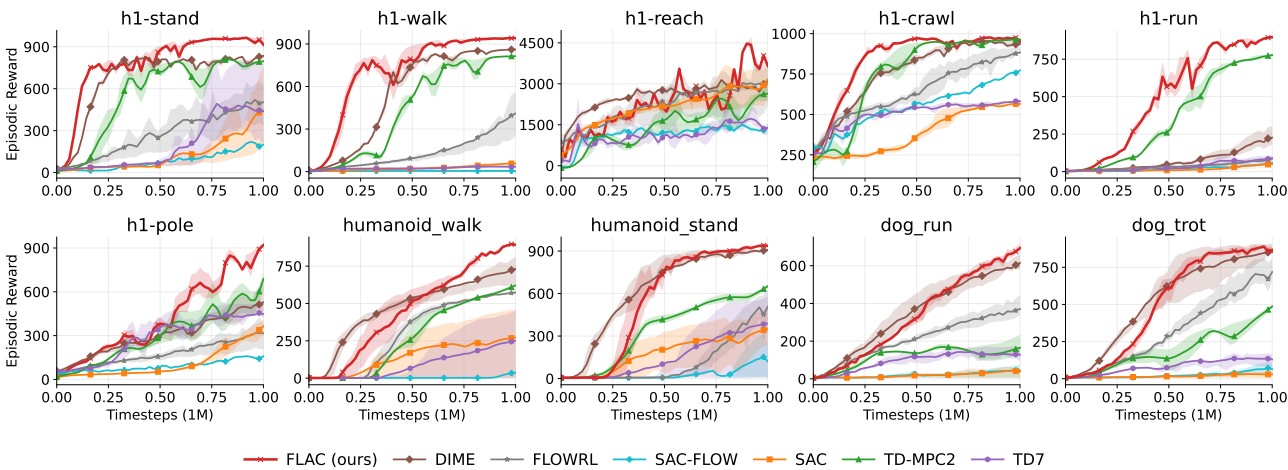

*Figure 2.* Main results. We provide performance comparisons on two challenging benchmarks. For comprehensive results, please refer to Appendix D. All model-free algorithms are evaluated with 5 random seeds, while the model-based algorithm (TD-MPC2) uses 3 seeds. DIME incorporates cross Q-learning (Simmons-Edler et al., 2019) to boost performance, whereas FLAC does not rely on these enhancements.

To address this, we formulate the energy regulation as a constrained optimization problem. Instead of manually tuning the penalty weight, we specify a target energy budget $E_{\text{tgt}}$, representing the desired level of stochasticity in the generation process. The objective is to maximize the expected return subject to the constraint that the average kinetic energy remains below this threshold:

$$\max_{\pi} \mathbb{E}_{s\sim\mathcal{D}, a\sim\pi}[Q^{\pi}(s, a)] \quad \text{s.t.} \quad \mathbb{E}_{s\sim\mathcal{D}}[\widehat{\mathcal{E}}_{\pi}(s)] \leq E_{\text{tgt}}. \tag{15}$$

We solve this constrained problem via the Lagrangian dual method. We construct the Lagrangian with respect to a learnable multiplier $\alpha \geq 0$:

$$\min_{\alpha\geq 0} \max_{\pi} \mathcal{L}(\pi, \alpha) = \mathbb{E}\left[Q^{\pi}(s, a) - \alpha(\widehat{\mathcal{E}}_{\pi}(s) - E_{\text{tgt}})\right]. \tag{16}$$

The optimization of the policy $\pi$ (Actor Update) corresponds to maximizing $\mathcal{L}$ with respect to $\pi$, which recovers the energy-regularized objective in Eq. (14). For the multiplier $\alpha$, we minimize the dual objective:

$$J(\alpha) = \mathbb{E}_{s\sim\mathcal{D}}\left[\alpha \cdot (E_{\text{tgt}} - \widehat{\mathcal{E}}_{\pi}(s))\right]. \tag{17}$$

In practice, to ensure positivity, we parameterize the multiplier as $\alpha = \exp(\log\alpha)$ and update the log-multiplier $\log\alpha$ via gradient descent:

$$\log\alpha \leftarrow \log\alpha - \beta \cdot \mathbb{E}_{s\sim\mathcal{B}}\left[E_{\text{tgt}} - \text{stopgrad}(\widehat{\mathcal{E}}_{\theta}(s))\right]. \tag{18}$$

where $\beta$ is the learning rate.

This mechanism functions as a dynamic regulator for policy stochasticity. When the policy becomes too deterministic, $\alpha$ increases, forcing the generation process to adhere more closely to the high-entropy prior. Conversely, when the policy is sufficiently stochastic, $\alpha$ decreases, allowing the agent to pursue aggressive, high-reward trajectories.

## 5. Experiment

To comprehensively evaluate the effectiveness and generality of **FLAC**, we conduct experiments on a diverse set of challenging tasks from DMControl (Tassa et al., 2018) and HumanoidBench (Sferrazza et al., 2024). These benchmarks encompass high-dimensional locomotion and human-like robot (Unitree H1) control tasks. Our evaluation aims to answer the following key questions:

1. **Q1:** How does FLAC compare against state-of-the-art model-free and model-based baselines in terms of sample efficiency and asymptotic performance on high-dimensional continuous control tasks?

2. **Q2:** Does the proposed kinetic energy regularization effectively regulate policy stochasticity and improve performance?

3. **Q3:** How sensitive is FLAC to its key hyperparameters, specifically the target energy budget, and does the automatic Lagrangian tuning mechanism outperform fixed regularization schemes?

We compare FLAC against two categories of strong baselines:

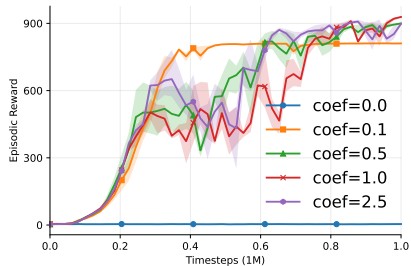
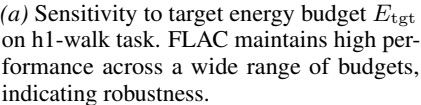
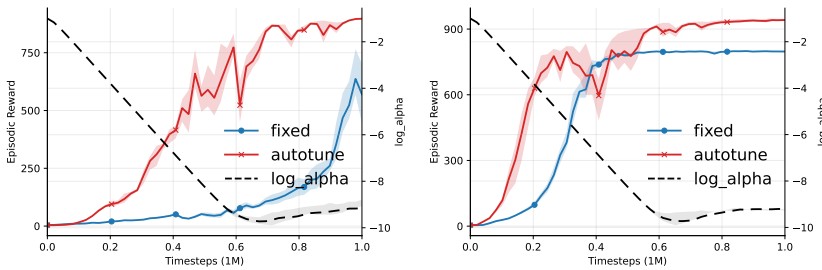

*(a)* Sensitivity to target energy budget $E_{\text{tgt}}$ on h1-walk task. FLAC maintains high performance across a wide range of budgets, indicating robustness.

*(b)* Efficacy of automatic Lagrangian tuning on h1-run (left) and h1-walk (right). Evolution of $\log \alpha$ during training shows a "decrease-then-increase" pattern, indicating that FLAC automatically relaxes constraints for early learning and tightens them later to enforce exploration.

*Figure 3.* Ablation Studies

- **Model-free RL:** We include deterministic policies (TD7 (Fujimoto et al., 2023)), standard Gaussian policies (SAC (Haarnoja et al., 2018)), and recent diffusion/flow-based methods (DIME (Celik et al., 2025), SAC-FLOW (Zhang et al., 2025), and FlowRL (Lv et al., 2025)).

- **Model-based RL:** We include TD-MPC2 (Hansen et al., 2023), a leading model-based algorithm across different benchmarks, to benchmark the asymptotic performance limits. Note that model-based methods are not directly comparable to model-free approaches due to differences in underlying assumptions and access to environment dynamics; TD-MPC2 is included as a reference for asymptotic performance.

### 5.1. Main Results

**Performance across Environments.** Figure 2 presents the comparative learning curves across diverse continuous control tasks. We observe that **FLAC** consistently matches or exceeds strong model-free baselines. This robustness extends to high-dimensional state spaces, specifically in the DMC Dog domain ($s \in \mathbb{R}^{223}, a \in \mathbb{R}^{38}$) and the contact-rich HumanoidBench Unitree H1 task. Furthermore, compared to the model-based benchmark TD-MPC2 (Hansen et al., 2023), FLAC attains comparable asymptotic returns, achieving this within a model-free framework that bypasses the need for world model learning or online planning.

**Comparison with Other Diffusion/Flow-based Policies.** When compared with prior diffusion-based and flow-based policies, FLAC demonstrates superior or comparable asymptotic performance relative to strong baselines such as DIME (Celik et al., 2025) and SAC-Flow (Zhang et al., 2025). FLAC attains these results using $N = 2$ number of function evaluations (NFE) per action throughout training and evaluation. In contrast, these baselines typically require more discretization steps to approximate the policy, with DIME using $N = 16$ and SAC-Flow using

$N = 4$. Moreover, DIME further benefits from cross Q-learning (Simmons-Edler et al., 2019) as an additional performance enhancement, whereas FLAC does not rely on this technique.

### 5.2. Ablation Studies

To rigorously verify the robustness and the internal mechanism of FLAC, we conduct two sets of ablation studies.

**Sensitivity to Target Energy Budget.** We first investigate the sensitivity of FLAC to the target energy budget $E_{\text{tgt}}$. As shown in Appendix E, under an isotropic action-generation prior the expected kinetic energy scales approximately linearly with the action dimension, motivating a dimension-normalized parametrization $E_{\text{tgt}} = \mathcal{C} \cdot \dim(\mathcal{A})$. We evaluate performance across a wide range of coefficients $\mathcal{C} \in \{0, 0.1, 0.5, 2.5\}$.

As shown in Figure 3a, FLAC exhibits robustness, maintaining high performance across a broad range of energy budgets. Significant performance degradation is observed when the budget is tight ($\mathcal{C} \in \{0, 0.1\}$). Specifically, the limiting case of $\mathcal{C} = 0$ corresponds to a vanishing kinetic energy budget. In this regime, the regulation mechanism strictly suppresses the learned velocity field, compelling the policy to be fully random. The resulting poor performance is theoretically expected and empirically validates the efficacy of our kinetic energy constraint, confirming that the mechanism effectively governs the deviation from the prior. Beyond this extreme regime, the exact value of $E_{\text{tgt}}$ is not critical, simplifying hyperparameter tuning.

**Efficacy and Dynamics of Automatic Tuning.** To understand the operational mechanism of FLAC's automatic tuning, we compare our adaptive Lagrangian adjustment with fixed regularization schemes. Figure 3b illustrates two main observations.

Regarding Performance Superiority, the adaptive method consistently outperforms the fixed setting. Static regulariza-

tion schemes often yield suboptimal outcomes: excessive penalization restricts the policy to the uninformative prior, thereby hindering reward maximization, whereas insufficient penalization fails to effectively regularize the generation dynamics, resulting in premature convergence or training instability. In contrast, the adaptive mechanism dynamically identifies the optimal trade-off throughout the learning process.

Furthermore, the evolution of the learnable multiplier $\log \alpha$ (shown in Figure 3b) reveals the inner workings of FLAC. We observe a distinct trend where $\log \alpha$ initially decreases and subsequently increases. During the early stages, the penalty decreases; this relaxation allows the agent to prioritize value maximization by reaching high-reward regions. In the later stages, however, $\log \alpha$ increases, tightening the kinetic energy constraint. By forcing the generation flow to maintain lower energy, the mechanism pulls the policy geometrically closer to the high-entropy prior. Consequently, this process actively enhances exploration as the policy converges, effectively preventing premature mode collapse.

This dynamic behavior firmly validates our hypothesis: the kinetic energy regularization serves as an active, state-aware regulator that automatically transitions the agent from aggressive learning to entropy-constrained convergence.

## 6. Related Work

### Iterative Generative Policies.

In offline RL and imitation learning, diffusion/flow policies serve as flexible behavior models or policy classes trained from fixed datasets, where mode coverage are central (Levine et al., 2020; Wang et al., 2022; Yang et al., 2023; Chi et al., 2023; Park et al., 2025). Recent work studies value-/energy-guided training and sampling, where Q-values or learned energies bias generators toward high-return actions while maintaining data support (Ding et al., 2024; Psenka et al., 2023; Lu et al., 2023; Jain et al., 2024). For online RL, iterative policies have begun to be combined with actor-critic updates and efficiency-oriented designs (Wang et al., 2024; Celik et al., 2025; Lv et al., 2025; Zhang et al., 2025). Beyond RL benchmarks, diffusion/flow policies are also used in robotics and visuomotor control as general action-generation modules, underscoring their practical scalability when coupled with strong representation learning (Chi et al., 2023).

### Entropy Regulators for Generative Policies.

Maximum-entropy RL encourages exploration via entropy or KL regularization (Haarnoja et al., 2018; Ziebart, 2010). However, for policies defined implicitly by iterative samplers (diffusion/flow), the induced action density may be unavailable, making density-based regularization expensive

or fragile in online RL with limited solver budgets. Likelihood evaluation can be tied to change-of-variables along ODE dynamics (Chen et al., 2018) or path marginalization in SDEs (Song et al., 2020), both of which are nontrivial in practice. Recent methods integrate iterative generative policies with off-policy actor–critic learning by introducing practical entropy/exploration regulators tailored to diffusion/flow samplers. DIME (Celik et al., 2025) optimizes a complex variational surrogate objective of entropy to control stochasticity. Wang et al. (Wang et al., 2024) approximate the policy entropy with a multivariate Gaussian and use it to calibrate exploration noise. Zhang et al. (Zhang et al., 2025) train an additional noise-estimation network to enable entropy-style regularization for flow policies.

### Schrödinger Bridges: Path-Space KL, Optimal Transport, and GSB.

Schrödinger bridges provide a variational formulation for the most likely stochastic evolution between distributions relative to a reference diffusion, linking entropy regularization, stochastic control, and optimal transport (Léonard, 2013). Deterministic limits recover Benamou–Brenier kinetic-energy optimal transport (Mikami, 2004; Benamou & Brenier, 2000), which also motivates transport-learning methods (Lipman et al., 2022; Liu et al., 2022). On the stochastic side, learning-based SB solvers and diffusion-SB connections have been developed for fitting stochastic transports (Pavon et al., 2021; Vargas et al., 2021; Shi et al., 2023). The generalized Schrödinger bridge further relaxes hard terminal constraints into soft terminal potentials, yielding one-ended objectives aligned with decision-making settings where targets are specified by utilities or rewards (Liu et al., 2024).

## 7. Conclusion

In this work, we introduced **Field Least-Energy Actor-Critic (FLAC)**, establishing a unified perspective that maps Reinforcement Learning onto the Generalized Schrödinger Bridge framework. We theoretically demonstrated that the Maximum Entropy principle naturally emerges from minimizing kinetic energy, which acts as a computable geometric proxy for bounding the divergence from the reference process without explicit density estimation. Empirically, FLAC demonstrates highly competitive performance against strong baselines. However, similar to standard maximum entropy approaches, our current framework applies an isotropic regularization across all action dimensions. This treats distinct actuators uniformly, leaving for future work in developing anisotropic or state-dependent energy constraints to better accommodate tasks where varying degrees of stochasticity are required across different control channels.

## Impact Statement

This paper presents an advancement in Reinforcement Learning by integrating efficient generative policies into continuous control. Our method, FLAC, enhances the capability and sample efficiency of autonomous systems in high-dimensional tasks, such as robotic manipulation and humanoid locomotion. While this offers significant potential for industrial automation and embodied AI, the deployment of such learning-based controllers in real-world settings carries inherent safety risks. Specifically, the stochastic nature of generative policies, if not properly constrained, could lead to unpredictable behaviors in safety-critical scenarios. Therefore, rigorous safety verification, sim-to-real transfer protocols, and fail-safe mechanisms are essential prerequisites before deploying these algorithms in physical environments. We do not foresee immediate negative societal impacts beyond the general considerations associated with the development of autonomous decision-making systems.

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

# A. Proofs in the Main Text

**Notation.** In this appendix, we denote the generic distance by $D(\cdot\|\cdot)$. We analyze the Kinetic Energy $\mathcal{E}(u) = \mathbb{E}[\int_0^1 \frac{1}{2}\|u_\tau\|^2 d\tau]$ in both stochastic and deterministic regimes.

**Technical Assumptions.** To ensure the well-posedness of the theoretical results, we make the following standard assumptions throughout the paper:

1. **Regularity of Drift:** The vector field $u_\theta(s, \tau, x)$ is Lipschitz continuous in $x$ and adapted to the filtration. It satisfies the Novikov condition $\mathbb{E}[\exp(\frac{1}{2\sigma^2} \int \|u\|^2 d\tau)] < \infty$, ensuring the validity of the Girsanov transformation.

2. **Boundedness:** The action space $\mathcal{A}$ is bounded (e.g., $[-1, 1]^d$), and the reward function $r(s, a)$ is bounded. The reference prior $\mu_0$ is uniform over $\mathcal{A}$.

3. **Absolute Continuity:** The policy distribution $\pi(\cdot|s)$ is absolutely continuous with respect to the reference prior $\mu_0$ (i.e., $\pi \ll \mu_0$), ensuring the KL divergence is well-defined.

## A.1. Stochastic Regime: Energy as KL Divergence

We derive the equivalence between KL divergence and Kinetic Energy for SDEs ($\sigma > 0$).

**Setup.** Let $\mathbb{P}_s^{\mathrm{ref}}$ be the reference path measure induced by $dX_\tau = \sigma dW_\tau$. Let $\mathbb{P}_s^\theta$ be the policy path measure induced by $dX_\tau = u_\theta d\tau + \sigma dW_\tau$. Both share the initial distribution $X_0 \sim \mu_0$.

**Derivation.** Define $\beta_\tau := \frac{1}{\sigma} u_\theta(s, \tau, X_\tau)$. By Girsanov's Theorem, the log-Radon-Nikodym derivative is:

$$\log \frac{d\mathbb{P}_s^\theta}{d\mathbb{P}_s^{\mathrm{ref}}} = \int_0^1 \beta_\tau^\top dW_\tau - \frac{1}{2} \int_0^1 \|\beta_\tau\|^2 d\tau. \tag{19}$$

Under the measure $\mathbb{P}_s^\theta$, we can rewrite $dW_\tau = d\widetilde{W}_\tau + \beta_\tau d\tau$, where $\widetilde{W}_\tau$ is a standard Brownian motion. Substituting this back:

$$\log \frac{d\mathbb{P}_s^\theta}{d\mathbb{P}_s^{\mathrm{ref}}} = \int_0^1 \beta_\tau^\top d\widetilde{W}_\tau + \frac{1}{2} \int_0^1 \|\beta_\tau\|^2 d\tau. \tag{20}$$

Taking the expectation $\mathbb{E}_{\mathbb{P}_s^\theta}$, the stochastic integral (martingale) term vanishes:

$$D_{\mathrm{KL}}(\mathbb{P}_s^\theta\|\mathbb{P}_s^{\mathrm{ref}}) = \mathbb{E}_{\mathbb{P}_s^\theta}\left[\frac{1}{2}\int_0^1 \|\beta_\tau\|^2 d\tau\right] = \frac{1}{\sigma^2}\mathcal{E}(u). \tag{21}$$

$\square$

## A.2. Deterministic Regime: Energy as Wasserstein-2 Distance

We show that in the ODE limit ($\sigma \to 0$), the Kinetic Energy bounds the Wasserstein-2 distance.

**Setup.** Consider the continuity equation describing the evolution of the probability density $\rho_\tau$ driven by the vector field $u_\tau$:

$$\partial_\tau \rho_\tau + \nabla \cdot (\rho_\tau u_\tau) = 0. \tag{22}$$

The Benamou-Brenier formula (Benamou & Brenier, 2000) states that the squared Wasserstein-2 distance between two distributions $\mu_0$ and $\mu_1$ is the infimum of the kinetic energy over all valid velocity fields transporting $\mu_0$ to $\mu_1$:

$$\mathcal{W}_2^2(\mu_0, \mu_1) = \inf_{(v, \rho)} \left\{ \int_0^1 \int_{\mathbb{R}^d} \|v(x, \tau)\|^2 \rho(x, \tau) dx d\tau \right\}, \tag{23}$$

subject to the continuity equation and boundary conditions $\rho_0 = \mu_0, \rho_1 = \mu_1$.

**Connection to FLAC.** Our learned policy $u_\theta$ generates a specific flow that transports $\mu_0$ to a terminal distribution $\pi_\theta = \rho_1$. By definition, the energy of our specific flow $\mathcal{E}(u_\theta)$ is one candidate in the set of all possible transport plans. Therefore, it serves as an upper bound on the optimal transport cost:

$$\mathcal{W}_2^2(\mu_0, \pi_\theta) \leq 2\mathcal{E}(u_\theta). \tag{24}$$

Minimizing $\mathcal{E}(u_\theta)$ thus minimizes the upper bound on the geometric distance between the prior $\mu_0$ and the policy $\pi_\theta$. Moreover, when $\mu_0$ is a uniform distribution, this objective is related to the maximum entropy objective which pushing policy close to a uniform distribution.

**Remark (ODE limit and entropy).** In the deterministic (ODE) limit, controlling the deviation from a uniform prior in $\mathcal{W}_2$ is a geometric proximity constraint and does not, in general, imply a large terminal (differential) entropy.

Nevertheless, in continuous-control RL the practical role of maximum-entropy regularization is often to prevent premature policy concentration and early commitment to suboptimal modes (i.e., poor local optima), by maintaining broadly supported stochastic action sampling and sustained exploration.

From this perspective, this energy/$\mathcal{W}_2$ regularization provides a useful surrogate: it penalizes aggressive, large-scale transport (high control effort), which empirically discourages rapid concentration of probability mass and promotes coverage of the bounded action domain.

Moreover, the theoretical constructions that decouple $\mathcal{W}_2$-proximity from distributional spread typically rely on extreme local volume compression, and are often associated with highly non-uniform Jacobians of the induced flow. In practice, such behaviors are less likely to be realized under our policy parameterization and training dynamics: neural networks trained with first-order methods exhibit an empirical bias toward smoother, low-complexity solutions (often referred to as spectral bias), and the resulting learned transports tend to remain relatively regular under our energy regularization. Accordingly, in the deterministic regime we view the energy/$\mathcal{W}_2$ constraint as a geometric inductive bias that empirically mitigates global collapse and encourages broadly supported action sampling, rather than as a strict information-theoretic bound.

$\square$

### A.3. Proof of Terminal Entropy Control

We prove that minimizing path divergence controls the terminal distribution.

Let $\Pi(X_{0:1}) = X_1$ be the projection to the terminal state. Let $\pi_\theta = \mathbb{P}_s^\theta \circ \Pi^{-1}$ and $\mu_1^{\mathrm{ref}} = \mathbb{P}_s^{\mathrm{ref}} \circ \Pi^{-1}$.

By the Data Processing Inequality (DPI) for f-divergences (including KL):

$$D_{\mathrm{KL}}(\pi_\theta \| \mu_1^{\mathrm{ref}}) \leq D_{\mathrm{KL}}(\mathbb{P}_s^\theta \| \mathbb{P}_s^{\mathrm{ref}}). \tag{25}$$

Combining this with the result from Appendix A.1, we have:

$$D_{\mathrm{KL}}(\pi_\theta \| \mu_1^{\mathrm{ref}}) \leq \frac{1}{\sigma^2} \mathcal{E}(s). \tag{26}$$

Thus, minimizing Kinetic Energy forces the terminal policy $\pi_\theta$ to remain close to the high-entropy prior $\mu_1^{\text{ref}}$. Moreover, when $\mu_1^{\text{ref}}$ is a uniform distribution, this objective is related to the maximum entropy objective.

$\square$

## A.4. Proof of Proposition 3.1 (Optimal GSB Solution)

**Proposition Restatement.** *The unique optimal path measure $\mathbb{P}^*$ that minimizes the One-Ended GSB objective (Eq. 8) induces a terminal marginal distribution $p^*(X_1)$ of the form:*

$$p^*(X_1) \propto p_{\text{ref}}(X_1) \cdot \exp\left(-\frac{\mathcal{G}(X_1)}{\alpha}\right).$$

*Proof.* The Generalized Schrödinger Bridge problem can be viewed as a static variational problem on the space of path measures. The objective function is:

$$\mathcal{J}(\mathbb{P}) = \alpha \mathcal{D}(\mathbb{P}\|\mathbb{P}^{\text{ref}}) + \mathbb{E}_\mathbb{P}[\mathcal{G}(X_1)]. \tag{27}$$

Recall that the KL divergence is defined as

$$\mathcal{D}(\mathbb{P} \mid \mathbb{Q}) = \int \log\left(\frac{d\mathbb{P}}{d\mathbb{Q}}\right) d\mathbb{P}.$$

Substituting this into the objective:

$$\mathcal{J}(\mathbb{P}) = \alpha \int \log\left(\frac{d\mathbb{P}}{d\mathbb{P}^{\text{ref}}}\right) d\mathbb{P} + \int \mathcal{G}(X_1) d\mathbb{P} \tag{28}$$

$$= \alpha \int \left[\log\left(\frac{d\mathbb{P}}{d\mathbb{P}^{\text{ref}}}\right) + \frac{\mathcal{G}(X_1)}{\alpha}\right] d\mathbb{P}. \tag{29}$$

Note that $\frac{\mathcal{G}(X_1)}{\alpha} = \log \exp\left(\frac{\mathcal{G}(X_1)}{\alpha}\right)$, thus:

$$\mathcal{J}(\mathbb{P}) = \alpha \int \log\left(\frac{d\mathbb{P}}{d\mathbb{P}^{\text{ref}}} \cdot \exp\left(\frac{\mathcal{G}(X_1)}{\alpha}\right)\right) d\mathbb{P}. \tag{30}$$

Define an unnormalized auxiliary measure $\tilde{\mathbb{Q}}$ such that

$$d\tilde{\mathbb{Q}} = \exp\left(-\frac{\mathcal{G}(X_1)}{\alpha}\right) d\mathbb{P}^{\text{ref}}.$$

Then the term inside the logarithm becomes $\frac{d\mathbb{P}}{d\tilde{\mathbb{Q}}}$. The objective is minimized when $\mathbb{P}$ matches the normalized version of $\tilde{\mathbb{Q}}$. Therefore, the optimal measure $\mathbb{P}^*$ satisfies:

$$\frac{d\mathbb{P}^*}{d\mathbb{P}^{\text{ref}}}(\omega) \propto \exp\left(-\frac{\mathcal{G}(X_1(\omega))}{\alpha}\right). \tag{31}$$

Marginalizing this path measure at $\tau = 1$, we obtain the terminal distribution:

$$p^*(X_1) = \frac{d\mathbb{P}_1^*}{dx}(x) \propto p_{\text{ref}}(X_1) \exp\left(-\frac{\mathcal{G}(X_1)}{\alpha}\right). \tag{32}$$

This concludes the proof. $\square$

## A.5. Proof of Proposition 4.1

Fix a policy $\pi$.

**Bellman operator.** Recall the energy-regularized Bellman evaluation operator:

$$(\mathcal{T}^\pi Q)(s, a) := r(s, a) + \gamma \, \mathbb{E}_{\substack{s' \sim p(\cdot|s,a) \\ a' \sim \pi(\cdot|s')}} \left[ Q(s', a') - \alpha \, \mathcal{E}_\pi(s') \right]. \tag{33}$$

Here $\mathcal{E}_\pi(s')$ denotes the expected kinetic energy required to sample $a' \sim \pi(\cdot \mid s')$.

**Contraction in $\| \cdot \|_\infty$.** For any two bounded functions $Q_1, Q_2$ and any $(s, a)$, we have

$$\left| (\mathcal{T}^\pi Q_1)(s, a) - (\mathcal{T}^\pi Q_2)(s, a) \right| = \gamma \left| \mathbb{E}_{s', a'} \left[ Q_1(s', a') - Q_2(s', a') \right] \right| \tag{34}$$

$$\leq \gamma \, \mathbb{E}_{s', a'} \left[ \left| Q_1(s', a') - Q_2(s', a') \right| \right] \tag{35}$$

$$\leq \gamma \| Q_1 - Q_2 \|_\infty, \tag{36}$$

where the expectations are over $s' \sim p(\cdot \mid s, a)$ and $a' \sim \pi(\cdot \mid s')$.

Taking the supremum over $(s, a)$ yields

$$\| \mathcal{T}^\pi Q_1 - \mathcal{T}^\pi Q_2 \|_\infty \leq \gamma \| Q_1 - Q_2 \|_\infty.$$

Thus $\mathcal{T}^\pi$ is a $\gamma$-contraction.

**Existence and uniqueness of the fixed point.** By fixed-point theorem, $\mathcal{T}^\pi$ has a unique fixed point $Q^\pi$.

**Identification with the regularized return.** Unrolling the fixed-point equation $Q^\pi = \mathcal{T}^\pi Q^\pi$ gives

$$Q^\pi(s, a) = \mathbb{E} \left[ r(s_0, a_0) + \gamma \left( Q^\pi(s_1, a_1) - \alpha \, \mathcal{E}_\pi(s_1) \right) \, \Big| \, s_0 = s, a_0 = a \right] \tag{37}$$

$$= \mathbb{E} \left[ \sum_{t \geq 0} \gamma^t r(s_t, a_t) \; - \; \alpha \sum_{t \geq 1} \gamma^t \mathcal{E}_\pi(s_t) \, \Big| \, s_0 = s, a_0 = a \right], \tag{38}$$

where $s_{t+1} \sim p(\cdot \mid s_t, a_t)$ and $a_{t+1} \sim \pi(\cdot \mid s_{t+1})$.

$$\square$$

## A.6. Proof of Proposition 4.2

Fix a policy $\pi$ and let $Q^\pi$ be the unique fixed point of $\mathcal{T}^\pi$ defined in Eq. (11) (i.e., $Q^\pi = \mathcal{T}^\pi Q^\pi$).

**Policy improvement condition.** Assume the updated policy $\pi_{\text{new}}$ satisfies, for all states $s$,

$$\mathbb{E}_{a \sim \pi_{\text{new}}(\cdot|s)}[Q^\pi(s, a)] - \alpha \, \mathcal{E}_{\pi_{\text{new}}}(s) \geq \mathbb{E}_{a \sim \pi(\cdot|s)}[Q^\pi(s, a)] - \alpha \, \mathcal{E}_\pi(s). \tag{39}$$

**Show one-step improvement in Bellman backup.** Consider the Bellman evaluation operators $\mathcal{T}^\pi$ and $\mathcal{T}^{\pi_{\text{new}}}$. For any $(s, a)$,

$$(\mathcal{T}^{\pi_{\text{new}}} Q^\pi)(s, a) = r(s, a) + \gamma \, \mathbb{E}_{s' \sim p(\cdot|s,a)} \mathbb{E}_{a' \sim \pi_{\text{new}}(\cdot|s')} \left[ Q^\pi(s', a') - \alpha \, \mathcal{E}_{\pi_{\text{new}}}(s') \right]. \tag{40}$$

Applying (39) at state $s'$ yields

$$\mathbb{E}_{a' \sim \pi_{\text{new}}(\cdot|s')}[Q^\pi(s', a')] - \alpha \, \mathcal{E}_{\pi_{\text{new}}}(s') \geq \mathbb{E}_{a' \sim \pi(\cdot|s')}[Q^\pi(s', a')] - \alpha \, \mathcal{E}_\pi(s').$$

Taking expectation over $s' \sim p(\cdot \mid s, a)$ and substituting back gives

$$(\mathcal{T}^{\pi_{\text{new}}} Q^\pi)(s, a) \geq r(s, a) + \gamma \, \mathbb{E}_{s' \sim p(\cdot|s,a)} \mathbb{E}_{a' \sim \pi(\cdot|s')} \left[ Q^\pi(s', a') - \alpha \, \mathcal{E}_\pi(s') \right] \tag{41}$$

$$= (\mathcal{T}^\pi Q^\pi)(s, a). \tag{42}$$

Since $Q^\pi$ is the fixed point of $\mathcal{T}^\pi$, we have $(\mathcal{T}^\pi Q^\pi)(s, a) = Q^\pi(s, a)$; therefore

$$(\mathcal{T}^{\pi_{\text{new}}} Q^\pi)(s, a) \geq Q^\pi(s, a), \qquad \forall (s, a). \tag{43}$$

**Monotone convergence to the fixed point.** The operator $\mathcal{T}^{\pi_{\text{new}}}$ is monotone: if $Q_1 \leq Q_2$ pointwise then $\mathcal{T}^{\pi_{\text{new}}} Q_1 \leq \mathcal{T}^{\pi_{\text{new}}} Q_2$ (the reward and energy terms do not depend on $Q$ and expectations preserve order).

Apply $\mathcal{T}^{\pi_{\text{new}}}$ iteratively to (43):

$$Q^\pi \leq \mathcal{T}^{\pi_{\text{new}}} Q^\pi \leq (\mathcal{T}^{\pi_{\text{new}}})^2 Q^\pi \leq \cdots.$$

By Proposition 4.1, $\mathcal{T}^{\pi_{\text{new}}}$ is a $\gamma$-contraction; hence the sequence converges in $\|\cdot\|_\infty$ to its unique fixed point $Q^{\pi_{\text{new}}}$. Taking limits yields

$$Q^{\pi_{\text{new}}}(s, a) \geq Q^\pi(s, a), \qquad \forall (s, a),$$

which proves monotonic improvement.

$\square$

*Table 1.* Hyperparameters

|  | Hyperparameter | Value |
|---|---|---|
| **Hyperparameters** | Optimizer | Adam |
| | Critic learning rate | $3 \times 10^{-4}$ |
| | Actor learning rate | $3 \times 10^{-4}$ |
| | Discount factor | 0.99 |
| | Batch Size | 256 |
| | Replay buffer size | $1 \times 10^6$ |
| | Target energy | 0.5*dim(A) |
| | NFE steps $N$ | 2 |
| | Solver | Midpoint Euler |
| **Value network** | Network hidden dim | 512 |
| | Network hidden layers | 3 |
| | Network activation function | gelu |
| **Policy network** | Network hidden dim | 512 |
| | Network hidden layers | 2 |
| | Network activation function | elu |

## B. Baselines

In our experiments, we have implemented SAC, TD7, DIME,SAC-FLOW and TD-MPC2 using their original code bases and official results.

- SAC (Haarnoja et al., 2018), we utilized the open-source PyTorch implementation, available at `https://github.com/pranz24/pytorch-soft-actor-critic`.

- TD7 (Fujimoto et al., 2023) was integrated into our experiments through its official codebase, accessible at `https://github.com/sfujim/TD7`.

- TD-MPC2 (Hansen et al., 2023) was employed with its official implementation from `https://github.com/nicklashansen/tdmpc2` and used their official results.

- SAC-FLOW (Zhang et al., 2025) was employed with its official implementation from `https://github.com/Elessar123/SAC-FLOW.git`

- DIME (Celik et al., 2025) was employed with its official implementation from `https://github.com/ALRhub/DIME.git` and used their official results.

- FlowRL (Lv et al., 2025) was employed with its official implementation from `https://github.com/bytedance/FlowRL`

## C. Environment Details

We validate our algorithm on the DMControl (Tassa et al., 2018) and HumanoidBench (Sferrazza et al., 2024), including the most challenging high-dimensional and Unitree H1 humanoid robot control tasks. On DMControl, we focus on the most challenging tasks(dog and humanoid domains). On HumanoidBench, we focus on tasks that do not require dexterous hands.

| Task | State dim | Action dim |
|------|-----------|------------|
| Humanoid Stand | 67 | 24 |
| Humanoid Run | 67 | 24 |
| Humanoid Walk | 67 | 24 |
| Dog Run | 223 | 38 |
| Dog Trot | 223 | 38 |
| Dog Stand | 223 | 38 |
| Dog Walk | 223 | 38 |

*Table 2.* Task dimensions for DMControl.

| Task | Observation dim | Action dim |
|------|-----------------|------------|
| H1 Crawl | 51 | 19 |
| H1 Hurdle | 51 | 19 |
| H1 Maze | 51 | 19 |
| H1 Pole | 51 | 19 |
| H1 Reach | 57 | 19 |
| H1 Run | 51 | 19 |
| H1 Sit Hard | 64 | 19 |
| H1 Sit Simple | 51 | 19 |
| H1 Slide | 51 | 19 |
| H1 Stair | 51 | 19 |
| H1 Stand | 51 | 19 |
| H1 Walk | 51 | 19 |

*Table 3.* Task dimensions for HumanoidBench.

## D. Toy Example Setup

We consider a 2D multi-goal bandit to illustrate the effect of least-action regularization. The action space is $\mathcal{A} = \mathbb{R}^2$, with 8 goal positions placed uniformly on a circle of radius 4:

$$g_k = \left( 4\cos\left(\frac{2\pi k}{8}\right), 4\sin\left(\frac{2\pi k}{8}\right) \right), \quad k = 0, 1, \ldots, 7. \tag{44}$$

The reward function is the maximum Gaussian bump over all goals:

$$r(a) = \max_k \exp\left( -\frac{\|a - g_k\|^2}{2} \right). \tag{45}$$

Both policies use a 2-layer MLP drift field with base distribution $\nu = \mathcal{N}(0, I)$ and $K = 24$ Euler steps. Without regularization, Naive Flow collapses to a single mode (1/8 coverage) while its kinetic energy explodes. FLAC maintains bounded energy via dual ascent and discovers all 8 goals (8/8 coverage), demonstrating that least-action regularization prevents mode collapse.

## E. Estimation of Target Kinetic Energy

The heuristic adjustment of the target kinetic energy $E_{\text{tgt}}$ in our **Adaptive Kinetic Budgeting** mechanism draws direct inspiration from the target entropy heuristic used in Soft Actor-Critic (SAC). In SAC, the target entropy is typically set to

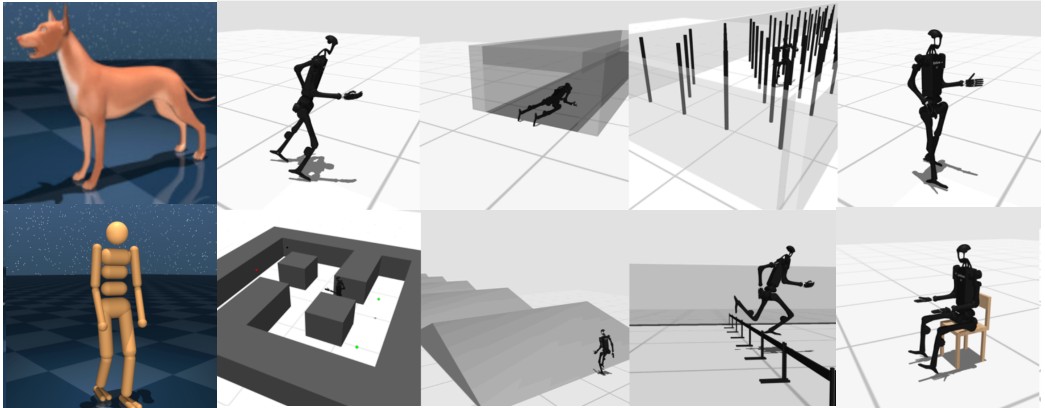

*Figure 4.* Task Domain Visualizations.

$\mathcal{H}_{\text{target}} = -\dim(\mathcal{A})$ to prevent the policy from collapsing into a deterministic point mass. Similarly, FLAC requires a reference value to regulate the trade-off between control effort and stochasticity. However, since we operate in the energy domain rather than entropy, we derive a geometric heuristic grounded in the physics of optimal transport.

Here, we derive a practical rule of thumb for setting $E_{\text{tgt}}$ based on the **Transport Cost** required to traverse the action space.

### E.1. Geometric Derivation

Consider a standard continuous control setting where the action space is bounded and normalized to $\mathcal{A} = [-1, 1]^d$. The generative policy evolves a latent state $X_\tau$ from a base distribution $X_0 \sim \mathcal{N}(0, I)$ (centered at the origin) to a terminal action $X_1$.

**Unit Displacement Cost.** Suppose the policy needs to generate an action at the boundary of the feasible space (e.g., $x = 1$) starting from the mean of the prior (e.g., $x = 0$). Under the Principle of Least Action, the most energy-efficient trajectory is a constant-velocity path (a geodesic):

$$u(\tau) = v, \quad \text{where } v = \frac{\Delta x}{\Delta \tau} = \frac{1 - 0}{1} = 1.$$

The kinetic energy consumed by this specific "unit" trajectory is:

$$\mathcal{E}_{\text{unit}} = \int_0^1 \frac{1}{2} \|u(\tau)\|^2 d\tau = \int_0^1 \frac{1}{2}(1)^2 d\tau = 0.5.$$

This implies that to deterministically shift the probability mass from the center to the boundary of the action space, the system must expend at least $0.5$ units of energy per dimension.

**Dimension Scaling.** Since the total kinetic energy is additive across independent dimensions (due to the squared norm $\|u\|^2 = \sum u_i^2$), the total energy required to reach the boundary in all $d$ dimensions is $0.5 \times d$.

### E.2. The Energy Budget Formula

Based on the derivation above, we formulate the target energy budget as a linear function of the action dimension:

$$E_{\text{tgt}} = \mathcal{C} \cdot \dim(\mathcal{A}), \tag{46}$$

where $\mathcal{C}$ is the **Energy Factor** representing the average allowable kinetic energy per dimension.

**Comparison with SAC.** In our experiments, we found that setting $\mathcal{C} \in [0.5, 2.5]$ yields robust performance across all tasks, and we set C=0.5, eliminating the need for per-task hyperparameter tuning. This offers a geometric counterpart to SAC's entropy heuristic.

**Robustness via Auto-tuning.** Crucially, the specific choice of $\mathcal{C}$ is not overly sensitive due to the **automatic tuning mechanism** of the Lagrange multiplier $\alpha$. The adaptive $\alpha$ dynamically scales the penalty weight to balance the energy constraint against the reward signal. Consequently, even if $\mathcal{C}$ is suboptimal, the algorithm can adjust $\alpha$ to find a stable equilibrium, making FLAC significantly less brittle than methods requiring fixed regularization weights.

# F. More Experimental Results

### F.1. Sensitivity to NFE

In all experiments, we set the number of function evaluations (NFE) to 2. We empirically observed that increasing NFE does help accelerate convergence in the early stages of training. However, it has little impact on the final performance as showed in Figure 5. This suggests that while higher NFE can facilitate faster initial learning, the ultimate effectiveness of the policy is not strongly dependent on this hyperparameter, the ultimate effectiveness of the policy is not strongly dependent on this hyperparameter. We hypothesize that this phenomenon arises because the kinetic-energy regularization biases the learned generation dynamics toward low-energy trajectories, which tend to be shorter and closer to straight-line transports from the prior to the action. This effect is also observed in the toy example (Figure 1), where energy regularization yields straighter and shorter transport paths.

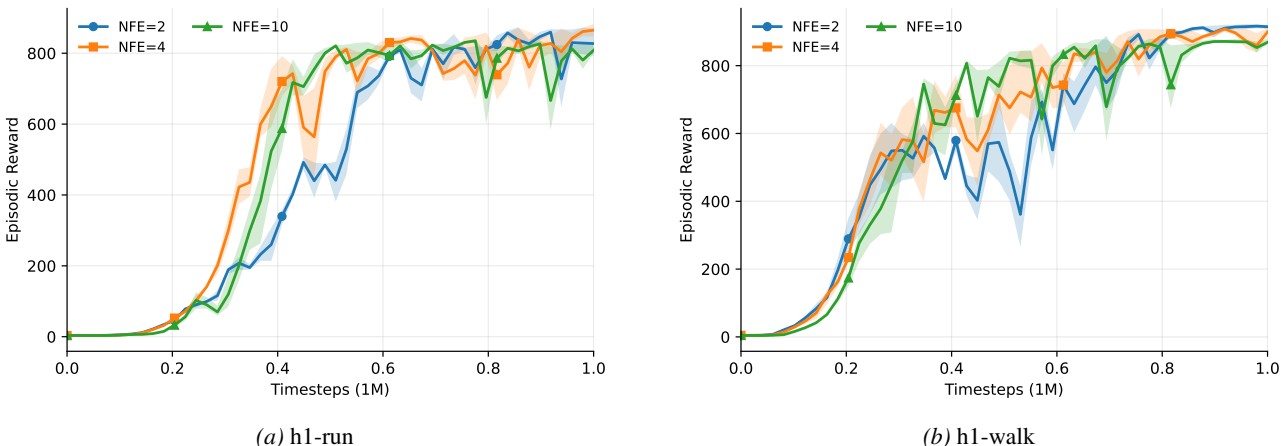

*(a)* h1-run                                                    *(b)* h1-walk

*Figure 5.* Sensitivity to NFE. Increasing NFE accelerates early convergence but has little impact on final performance.

This finding supports that, for FLAC: the use of a small, fixed NFE for efficient training without sacrificing the quality of the final results.

### F.2. Efficiency

In addition to sample efficiency, we also analyzed the overall computational efficiency of our algorithm in Figure 6. Specifically, we conducted a comparative study against DIME on seven challenging tasks from the DMC-hard benchmark. In these experiments, the horizontal axis represents wall-clock time. Although our implementation is based on PyTorch(with torch.compile for acceleration), thanks to the robustness of our method with respect to the NFE hyperparameter, our approach remains more efficient than DIME (failed to learn effectively at NFE=2), which is implemented in JAX. This demonstrates that our method achieves superior computational efficiency despite the differences in underlying frameworks.

### F.3. Comprehensive Results

We report the complete results on DMC-Hard and HumanoidBench in Fig. 8 and Fig. 7, respectively. On HumanoidBench, FLAC matches or outperforms all baselines on most tasks, while underperforming a strong model-based baseline on a small subset of tasks; on DMC-Hard, FLAC matches or outperforms all baselines across tasks.

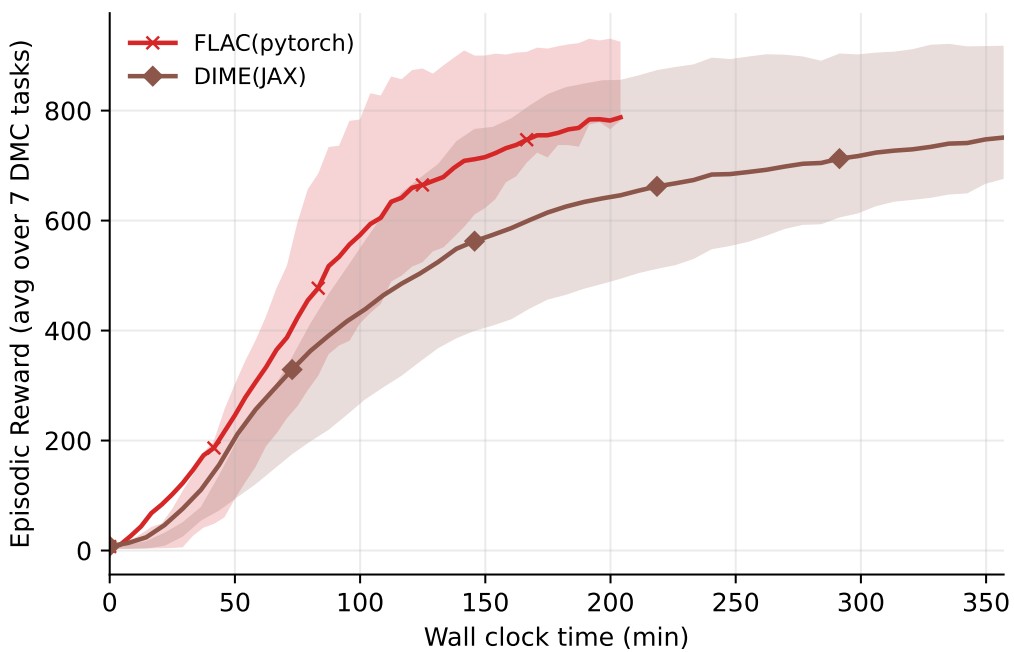

*Figure 6.* Computational Efficiency Comparison to DIME

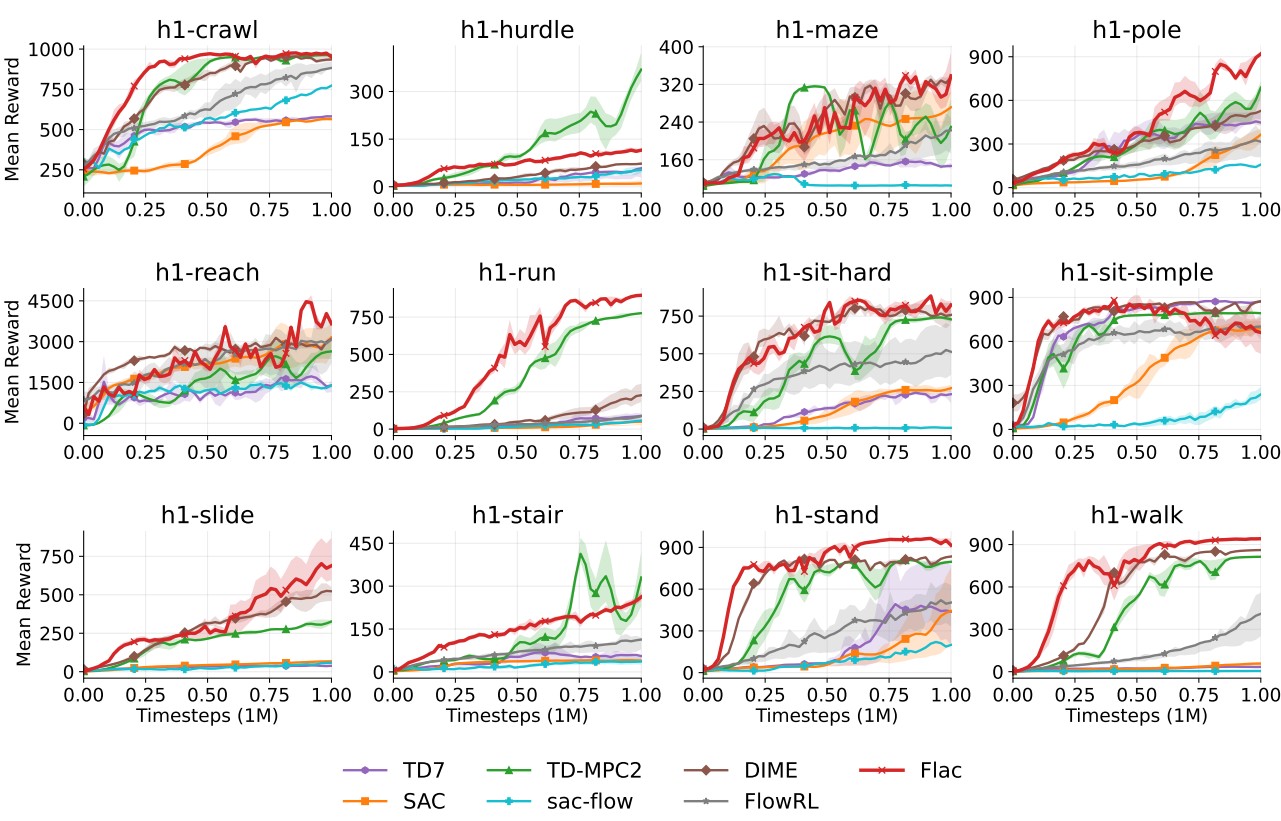

*Figure 7.* Full Results on Humanoid Bench.

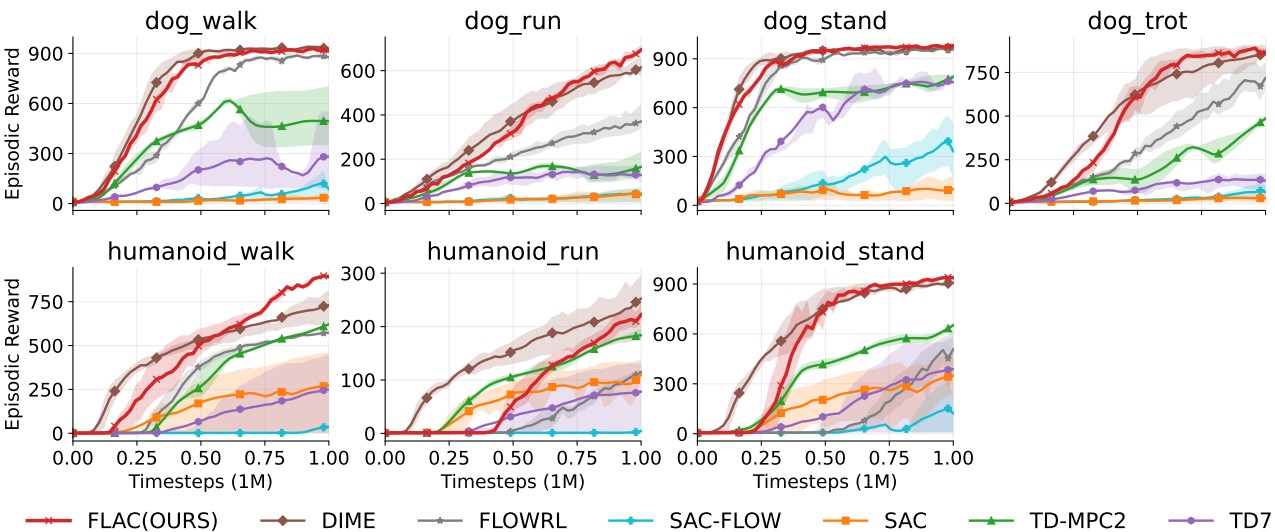

*Figure 8.* Full Results on DMC-Hard.

