# OpenReview forum: "FLAC: Maximum Entropy RL via Kinetic Energy Regularized Bridge Matching"
_ICML.cc/2026/Conference — ICML 2026 regular_

### Official Review · Reviewer_DyZM · 2026-03-10

**Soundness:** 3
**Presentation:** 3
**Significance:** 2
**Originality:** 2
**Overall Recommendation:** 4
**Confidence:** 2

**Summary:**

The paper proposes Field Least-Energy Actor-Critic (FLAC) to effectively integrate iterative generative policies with Maximum Entropy RL. Because iterative generators lack accessible action log-densities, the authors cleverly reframe policy optimization as a Generalized Schrödinger Bridge (GSB) problem. FLAC regulates policy stochasticity by penalizing the kinetic energy of the velocity field, acting as a proxy for divergence from a high-entropy reference. The method includes an automatic Lagrangian tuning mechanism and demonstrates strong, computationally efficient empirical performance.

**Compliance With Llm Reviewing Policy:**

Affirmed.

**Key Questions For Authors:**

1. Did you observe any empirical edge cases where the Wasserstein-2 geometric bias failed to prevent mode collapse, compared to traditional strict entropy regularization?
2. FLAC uses a constant NFE=2 during training and evaluation. Did you experiment with dynamic NFE scheduling? Could it potentially improve sample efficiency in the early learning phase?
3. The target energy budget is isotropic, defined as $E_{tgt}=\mathcal{C} \cdot dim(\mathcal{A})$. How sensitive is the framework to environments where different actuators require vastly different variances, and would an anisotropic constraint be mathematically tractable?

**Limitations:**

yes

**Strengths And Weaknesses:**

Soundness:Solid theoretical grounding, rigorously establishing the equivalence between KL divergence and Kinetic Energy for SDEs via Girsanov's Theorem.In the deterministic ODE limit, kinetic energy bounds the Wasserstein-2 distance. This acts as a geometric proximity constraint rather than a strict information-theoretic entropy guarantee, slightly weakening the "maximum entropy" claim in this specific regime.
Presentation:The narrative flows logically from abstract SBP/GSB concepts to a tractable RL algorithm, aided by a highly intuitive 2D toy example.The distinction between the stochastic and deterministic regimes—and specifically how the reference marginal changes could be introduced more clearly in the main text.
Significance: Offers a highly practical and computationally efficient solution (using only NFE=2) for generative RL without the need for auxiliary density estimation networks.Asymptotic performance gains over concurrent strong baselines (like DIME and SAC-FLOW) are sometimes comparable rather than strictly dominant.
Originality: Casting RL's entropy-regularized exploration as a least-kinetic energy transport problem is a highly creative synthesis of optimal transport and reinforcement learning.

---

> ### Author Rebuttal · Authors · 2026-03-31
>
> We sincerely thank the reviewer for the thoughtful evaluation and for highlighting both the strengths and the current boundaries of the paper. We especially appreciate your comments on the distinction between the stochastic and deterministic regimes, as well as your questions on edge cases and practical extensions.
>
> >On the stochastic and deterministic regimes
>
> Thank you for pointing out this subtle theoretical distinction. As discussed in the paper, we distinguish the stochastic SDE regime from the deterministic ODE regime, and we would like to clarify this distinction more explicitly here.
>
> In the **SDE regime**, Girsanov's theorem yields a path-space KL characterization, and Eq. (6) connects the kinetic energy term to the MaxEnt interpretation through the reference process. In the **ODE regime**, the corresponding control quantity becomes a geometric transport cost, leading to a Wasserstein-type proximity interpretation.
>
> While the formal interpretation differs across the two regimes, the loss itself is unified across the two regimes, and both serve the same practical regularization purpose: discouraging premature concentration of the policy and helping preserve exploration. From this perspective, the ODE formulation provides a transport-based realization of the same regularization purpose as **max-entropy-style regularization**, with the distinction lying in the geometric characterization rather than in its functional effect. We will revise the presentation to make this distinction, including the corresponding change in the role of the reference marginal, more explicit.
>
>
> > Q1: Edge cases under Wasserstein-type control
>
> This is an excellent question. In the deterministic regime, the regularizer acts through geometric proximity rather than directly through a terminal entropy quantity. At the same time, its regularization purpose remains closely aligned with that of entropy regularization: it discourages premature concentration and helps maintain useful action diversity during training.
>
> Across our experiments, we did **not** observe harmful collapse under FLAC. Figure 1 shows a challenging multimodal setting, where FLAC successfully recovers all modes. More broadly, on tasks such as **H1-walk**(Figure 7), where several methods have largely saturated, FLAC continues to improve even in the later stage of training, which is consistent with the view that the regularizer remains effective at maintaining useful action diversity during training.
>
> Although the deterministic interpretation is geometric rather than entropic, it was sufficient in our experiments to prevent the kind of excessive concentration that would be problematic for RL. We will clarify this point more carefully in the revision.
>
> > Q2: Dynamic NFE scheduling
>
> Thank you for this suggestion. Dynamic NFE scheduling is an interesting extension. However, increasing NFE directly increases inference cost, since each additional function evaluation requires another pass through the policy network. In addition, making the number of steps dynamic typically requires extra machinery to determine the schedule, and care is needed to keep the training and inference procedures well aligned.
>
> For these reasons, we chose to use a fixed NFE. Using a fixed number of function evaluations is a common and standard practice in generative policy, and it provides a simple and efficient default. We will mention dynamic NFE scheduling more explicitly as a possible future direction.
>
> > Q3: Anisotropic energy budget
>
> This is a very good point, and we agree it is an important limitation of the current formulation. As noted in the paper's **Limitations** section, the present version of FLAC still inherits, to some extent, the same simplifying assumption as SAC-style isotropic stochasticity control, namely that all action dimensions are regularized with a shared scalar budget.
>
> At the same time, the kinetic-energy formulation makes this extension quite natural. One can replace the scalar coefficient with a **vector-valued coefficient**, so that each action dimension has its own regularization strength and corresponding dual variable. In this sense, an anisotropic version is mathematically tractable. We therefore view this as a promising direction for future work, especially for systems with heterogeneous actuators, where dimension-wise control of stochasticity may be beneficial.
>
> We thank the reviewer again for the careful and constructive comments. They helped us better articulate both the strengths and the current boundaries of FLAC, and we will revise the paper accordingly.

---

> > ### Author Rebuttal · Reviewer_DyZM · 2026-04-02
> >
> > The rebuttal meaningfully strengthens the paper, particularly by highlighting the empirical evidence on preventing mode collapse and justifying the practical efficiency of a fixed NFE=2, which really help address my concerns. Regarding the anisotropic energy budget, I maintain my view that it is a crucial consideration.  Overall, I keep my positive rating.

---

> > > ### Author Response · Authors · 2026-04-03
> > >
> > > Thank you very much for the thoughtful follow-up and for your encouraging assessment. We are very glad that our rebuttal was helpful, and we appreciate the opportunity to further clarify the role of anisotropic energy budgeting in our framework.
> > >
> > > We especially appreciate your emphasis on the anisotropic energy budget, which is a very careful and insightful observation. In the current paper, our primary goal is to establish the connection between the Schrödinger Bridge perspective and MaxEnt RL, rather than to fully explore anisotropic stochasticity control. To the best of our knowledge, this aspect has also remained relatively underexplored in prior work (e.g., SAC-Flow, DIME, DACER).
> > >
> > > Empirically, FLAC still performs strongly on challenging tasks such as H1-walk, where the different roles of arms and legs naturally introduce substantial actuator heterogeneity. This suggests that, on the current benchmarks, the isotropic budget is not the main limiting factor. At the same time, as noted in our limitations and future work, the geometric form of our objective makes anisotropic extensions quite natural, and we agree that this is an important and interesting direction for future study.
> > >
> > > Thank you again for the constructive feedback and continued support.

---

### Official Review · Reviewer_Gea5 · 2026-03-12

**Soundness:** 4
**Presentation:** 4
**Significance:** 3
**Originality:** 3
**Overall Recommendation:** 5
**Confidence:** 5

**Summary:**

This paper proposed a method that optimizes a process-based policy with maximum entropy principles. The key insight of this paper is casting the policy optimization as a generalized Schrodinger Bridge problem, thereby converting the entropy constraint on the final action probabilities to minimizing the kinetic energy along the process, which is tractable to compute and optimize. Based on this understanding, this paper proposed a practical actor-critic style algorithm called FLAC, and provided a rigorous analysis of the policy iterations. Empirical experiments are conducted on DMControl dog tasks and Humanoid-Bench, which represent a class of difficult tasks in continuous locomotion tasks. FLAC outperforms all other methods, including model-based RL algorithms and other diffusion-based methods, demonstrating its strong performance and potential.

**Compliance With Llm Reviewing Policy:**

Affirmed.

**Final Justification:**

My concerns have been adequately addressed.

**Key Questions For Authors:**

+ From Equation 2 and Equation 5, it seems FLAC is using a fixed diffusion coefficient $\sigma$ for the reference SDE and the policy. This is in contast to modern diffusion model-based methods, which uses annealed time-dependent coefficients. Did the authors ablate this design?
+ According to Appendix A, the authors are using Girsanov theorem justify the introduced Kinetic Energy as the KL divergence between the path measures. However, the Girsanov theorem also applies to two diffusion processes with shared diffusion coefficients; you can easily define the optimization problem as maximizing the terminal potential while simultaneously minimizing the divergence w.r.t a reference reverse-time Brownian motion, and the algorithms and the optimal solution are the same. In this sense, why introducing the notion of Generalized Schrodinger Bridge?
+ In the sense of my last question, the difference between DIME and FLAC essentially lies in different regularization. DIME uses entropy as the regularization, while FLAC uses the KL divergence w.r.t a Brownian motion. I would expect more experiments or arguments to demonstrate the effect and differences between these two choices, with all other factors (critic learning method, network architectures, diffusion steps, and etc) controlled and fixed. Intuitively speaking, if we interpret entropy as KL divergence w.r.t the uniform distribution over action space, the approach used in this paper imposes a Gaussian prior on action space and this might have some effects.
+ What's the network architecture of the policy network and critic network? Or more specifically, are the policy and the critics using standard MLP, or advanced architectures like SimBa/BroNet? Besides, what is the parameterization of the critics? Are they scalar or distributional?
+ In figure 6, why is pytorch-based FLAC even faster than JAX-based DIME? Is it because of the different NFEs?
+ Some paper that I think should be included, because they all propose to use divergence between path measures from Girsanov theorem to characterize the divergence between action distributions.

[1] MINDE: Mutual information neural diffusion estimation. ICLR 2024.

[2] Behavior-regularized diffusion policy optimization for offline reinforcement learning. ICML 2025.

[3] On Flow Matching KL Divergence. arXiv preprint arXiv:2511.0548.

**Limitations:**

yes

**Strengths And Weaknesses:**

Strengths:
+ The empirical performance looks strong. Derivation of the proposed method is solid. Presentation of the method is clear and easy to follow.

Weaknesses: Please check my questions.

---

> ### Author Rebuttal · Authors · 2026-03-30
>
> We sincerely thank the reviewer for the detailed and technically precise feedback. Your comments helped us clarify several points that deserve better presentation, and we will revise the paper accordingly.
>
> > Q1: Fixed diffusion coefficient and annealed coefficients
>
> Thank you for raising this point. Our theory does **not** fundamentally rely on a fixed diffusion coefficient; we use constant $\sigma$ only for simplicity. The derivation extends to time-dependent $\sigma(\tau)$ with standard modifications to the control cost. We have added a comparison across different noise schedules; the results are available :https://anonymous.4open.science/r/FLACfigure/noise_schedules_compare.pdf. In practice, the scaling of $\sigma$ is less critical because much of its effect is absorbed by the automatically tuned dual variable $\alpha$ in the policy loss. We will clarify this point in the revision.
>
> > Q2: Why introduce a Generalized Schrödinger Bridge problem?
>
> Thank you for this insightful question. We agree that, when the controlled and reference processes share the same diffusion coefficient, the kinetic-energy regularizer can be derived directly from Girsanov's theorem. We also agree with your formulation. In fact, this is exactly the defining form of a one-ended GSB problem.
>
> We adopt the one-ended Generalized Schrödinger Bridge perspective because it is naturally aligned with the free-end structure of RL. First, RL is naturally a free-end problem: the initial distribution is known, while the terminal action distribution is optimized. This naturally corresponds to the one-ended bridge setting. Second, GSB gives the terminal characterization we need: Proposition 1 yields, which directly connects the bridge objective to the Boltzmann policy in MaxEnt RL. Girsanov gives the path-space KL, but not this terminal optimality characterization. Third, GSB explains why kinetic energy is the natural control cost in the underlying path space, rather than an ad hoc surrogate. In this sense, Girsanov is the technical tool, while GSB is the conceptual framework.
>
> We thank you again for this helpful formulation and will make this connection more explicit in the revision.
>
> > Q3/Q4: Controlled comparison with DIME, architecture details, and Gaussian prior effect
>
> We agree this is the right comparison. To isolate the effect, we added a controlled comparison that removes DIME-specific components (e.g., cross-Q learning) and follows the same setup reported in our hyperparameter table in the appendix. We matched the **policy architecture, critic architecture, NFE, and random seeds**. The results are available https://anonymous.4open.science/r/FLACfigure/humanoid_bench_flac_vs_dime.pdf.
>
> Under this controlled setting, FLAC still performs better than DIME. At small NFE, kinetic regularization remains effective because it imposes a direct stepwise constraint on the forward process, whereas DIME may be more sensitive to entropy-estimation quality in the low-step regime.
> In these experiments, both the policy and critics use standard MLP, and the critics are optimized by standard clipped double Q-learning. We did not use stronger backbones such as **SimBa** or **BroNet**, because our goal is to isolate the algorithmic effect of the proposed regularization. We will state this more explicitly in the paper.
>
> We also appreciate the concern about a possible Gaussian prior effect. FLAC does **not** require a Gaussian prior on the action space. The key requirement is that the reference terminal marginal $\mu_1^{\mathrm{ref}}$ be approximately **uniform** so that Proposition 1 recovers the Boltzmann policy. In practice, one may simply choose a uniform starting distribution, so the argument is not tied to Gaussianity. We will revise the wording accordingly.
>
> > Q5: Why is PyTorch FLAC faster than JAX DIME?
>
> Two factors contribute here. First, FLAC uses much smaller NFE than DIME, and each extra function evaluation incurs another forward and backward pass through the policy network. Second, our PyTorch implementation benefits from `torch.compile`, which further improves runtime.
>
> > Q6: Missing related work on path-measure divergence
>
> Thank you for these references. We agree they are relevant, especially since they also use path-measure divergence derived via Girsanov-type arguments to connect process-level and action-level distributions. We will add them and will discuss them more clearly in the revised version.
>
> We thank the reviewer again for the careful and constructive comments. They have helped us substantially improve the way we explain the motivation, scope, and technical positioning of the paper.

---

> > ### Author Rebuttal · Reviewer_Gea5 · 2026-04-03
> >
> > With respect to my Q3, since FLAC assumes the Brownian motion as the reference process (and since the marginal distribution of Brownian motion is still Gaussian), isn't it effectively applying a gaussian prior on the action space?
> >
> > In the ablation with DIME, are the noise schedule controlled? DIME also mentioned that they are using learnable noise schedules, could you control that as identical to the setup of FLAC?

---

> > > ### Author Response · Authors · 2026-04-03
> > >
> > > Thank you for this follow-up，it raises a precise point about the terminal distribution under the Brownian reference, and gives us the opportunity to clarify our formulation.
> > >
> > > On your first point, under a Brownian reference, the induced terminal marginal can indeed exhibit Gaussian smoothing. For example, if
> > > $$
> > > X_0 \sim \mathrm{Unif}([-a,a])  , dX_t = \sigma\ dW_t,
> > > $$
> > > and
> > > $$
> > > X_T = X_0 + \sigma W_T,
> > > $$
> > > then the terminal marginal is the convolution of the uniform law with a Gaussian kernel, with density
> > > $$
> > > p_T(x)=\frac{1}{2a}\left[\Phi\left(\frac{x+a}{\sigma\sqrt{T}}\right)-\Phi\left(\frac{x-a}{\sigma\sqrt{T}}\right)\right].
> > > $$
> > > This is closely related to the effect you pointed out, and we appreciate this careful observation. It also suggests that this aspect deserves a clearer discussion than our original text provided.
> > > In our formulation, however, Proposition 4.1 requires the terminal reference distribution over the bounded action domain to be sufficiently uniform. In the present Brownian setting, mild Gaussian smoothing remains compatible with this requirement, as an appropriate choice of reference scale keeps the terminal marginal close to uniform over the bounded action range.
> > >
> > > We adopt the zero-drift Brownian reference primarily for its simplicity and tractability: it leads directly to the kinetic-energy regularizer through Girsanov's theorem, without requiring more complex reference processes.
> > >
> > > To make this point more transparent, we additionally visualize particles initialized from a uniform distribution and evolved under Brownian motion, together with the resulting terminal marginal:
> > >
> > > https://anonymous.4open.science/r/FLACfigure/dist.md
> > >
> > > The empirical statistics are summarized below.
> > >
> > > | Distribution | Mean | Std | Mass in \([-1,1]\) | 5% Quantile | 95% Quantile |
> > > |---|---:|---:|---:|---:|---:|
> > > | Initial (uniform) | -0.000881 | 0.577709 | 1.000000 | -0.899967 | 0.899686 |
> > > | Terminal | -0.000867 | 0.577813 | 0.996103 | -0.900149 | 0.899693 |
> > >
> > > This example shows that, although Brownian evolution introduces Gaussian smoothing in principle, the terminal marginal can remain very close to uniform over the bounded action domain.
> > >
> > > Regarding the DIME comparison, following your Q3 suggestion, we fixed all components other than the regularization form so as to isolate its effect as cleanly as possible. Concretely, we disabled DIME's learnable noise schedule by fixing the friction coefficient (i.e., setting `learn_friction=False` in DIME's configuration).
> > >
> > > We hope this addresses your concerns, and we thank you again for the careful questions that prompted this clarification.

---

### Official Review · Reviewer_oZPy · 2026-03-13

**Soundness:** 4
**Presentation:** 3
**Significance:** 3
**Originality:** 4
**Overall Recommendation:** 5
**Confidence:** 3

**Summary:**

This paper presents an approach for training diffusion policies via maximum entropy RL. They cast the problem as a Generalized Schrodinger Bridge problem. Their formulation allows maximum entropy to emerge when the prior is a uniform distribution. The paper empirically evaluates their method on continuous control tasks and demonstrates  improvement over the baselines on a variety of tasks.

**Compliance With Llm Reviewing Policy:**

Affirmed.

**Final Justification:**

The rebuttal addressed my concerns, and I will maintain my positive score.

**Key Questions For Authors:**

1. Is this paper's approach compatible in the offline-to-online setting?
2. Is this paper's approach adaptable and useful in the setting where RL is started with a pretrained base policy (e.g. a diffusion policy)?

**Limitations:**

yes

**Strengths And Weaknesses:**

1. The paper adapts the Schrodinger Bridge Matching formulation of [4] to the RL setting. While the paper relies heavily on the work of [4], it is an interesting, novel application with strong empirical results.
2. The paper’s theory relies on a uniform prior, but in practice, RL is rarely done from scratch anymore. Instead RL typically starts with a pretrained policy. Their theory does not address such a setting, and they do not show experimental results for this setting either.
3. Additionally, their method still requires backpropagating through the generation process, similar to [1-3]. The paper mentions using fewer function evaluations for FLAC, but I am unclear whether fewer function evaluations simply “worked for this setting” or is a justified benefit of their approach. For example, one-step policies seem sufficiently expressive to perform well across a number of control tasks [5, 6].
4. The paper does not describe a search over the entropy parameters of baselines (e.g. SAC, SAC-FLOW, etc), while it does acknowledge a search over the kinetic energy parameter of FLAC. FLAC seems reasonably robust against varying kinetic energy parameters.

- [1] “Entropy-regularized Diffusion Policy with Q-Ensembles for Offline Reinforcement Learning”
- [2] “Diffusion Policies as an Expressive Policy Class for Offline Reinforcement Learning”
- [3] “Scaling Offline RL via Efficient and Expressive Shortcut Models”
- [4] “Generalized Schrödinger Bridge Matching”
- [5] “Flow Q-Learning”
- [6] “Consistency Models as a Rich and Efficient Policy Class for Reinforcement Learning”

---

> ### Author Rebuttal · Authors · 2026-03-30
>
> We sincerely thank the reviewer for the positive and constructive evaluation. We are encouraged that you found the paper technically solid, and we appreciate the opportunity to clarify your concerns.
>
> > W1/Q1/Q2: Uniform prior and pretrained/offline settings
>
> Thank you for raising this practical point. We have added **offline-to-online experiments**, where the policy is initialized from offline training and then **fine-tuned online**. The results are available: https://anonymous.4open.science/r/FLACfigure/off2on.md
>
> In our theory, the approximately uniform reference terminal marginal is used to recover the **Boltzmann policy required by standard MaxEnt RL**. This concerns the target regularization form rather than the data source. Offline data therefore do not remove the need for an entropy-style regularizer during online RL.
> This is also consistent with prior practice. Methods such as **RLPD, WSRL, and HilSERL** leverage offline or demonstration data, yet still retain entropy regularization or closely related exploration control during online improvement. From this perspective, FLAC's kinetic energy regularization serves as a principled alternative in the same setting.
>
> We also agree that fine-tuning from a pretrained policy is a natural extension. If the goal is stable adaptation around a pretrained base policy, one can replace the reference process $P_{\mathrm{ref}}$ with the process induced by that policy, equivalently by using the pretrained policy's velocity field as the reference drift. The kinetic energy term then penalizes deviation from the pretrained behavior, yielding a behavior-regularized fine-tuning effect.
>
> Thus, the GSB formulation suggests a natural extension to both settings: standard MaxEnt RL from scratch, and regularized adaptation from an existing policy. In this paper, we focus on the canonical MaxEnt RL setting, while the broader pretrained-policy setting remains a promising extension for future study.
>
> > W2: Low NFE, BPTT, and one-step policies
>
> We appreciate these closely related questions. As hypothesized in the paper, the strong performance at low NFE is related to kinetic energy regularization, which encourages shorter and straighter transport paths. This intuition is visible in the toy example and aligns with our theoretical motivation. Empirically, competing iterative-policy methods degrade much more severely in the **low-NFE regime**, whereas FLAC remains effective.
> This trend is also consistent with Appendix F.1: increasing NFE mainly improves early convergence but has limited effect on FLAC's final performance. The precise mechanism behind this effect is worth further study in future work.
>
> FLAC is optimized with **reparameterization-based pathwise gradients** through the generation process, so sequential backpropagation remains part of training, as in other iterative-policy methods. We agree this is an important practical consideration. That said, it is largely **orthogonal** to the main contribution of the paper: the MaxEnt RL formulation and regularization principle for implicit iterative policies. This cost can be reduced by standard choices such as **GRU or Transformer-based backbones**, similar in spirit to methods such as SAC-Flow. In this paper, we intentionally used the most basic **MLP**  to keep the comparison fair and isolate the effect of kinetic energy regularization.
>
> We also agree that one-step policies are a compelling direction. However, we view them as **complementary** rather than contradictory to FLAC. In many cases, one-step policies rely on an additional **distillation, shortcut, or consistency-style objective** to compress a stronger multi-step generator into a single step. Their strong performance therefore does not remove the relevance of iterative policies; rather, it builds on them through an extra compression method. FLAC studies the **training principle** for iterative generative policies under MaxEnt RL, while one-step methods study how to further **compress or accelerate** such policies. The same distillation techniques can in principle also be applied on top of a FLAC-trained policy if single-step inference is desired.
>
>
> > W3: Entropy tuning of baselines
>
> Thank you for pointing this out. All baselines were run using their **official codebases and recommended hyperparameters**. In particular, SAC and SAC-based variants use their tuned entropy-tuning procedures from the original implementations. For FLAC, we reported sensitivity to the kinetic energy parameter because it is method-specific and central to our formulation. We will clarify this protocol more explicitly in the final version.
>
> We thank the reviewer again for the supportive assessment and the constructive suggestions. We hope these clarifications make the scope of the current paper and the broader applicability of the GSB formulation clearer.

---

> > ### Author Rebuttal · Reviewer_oZPy · 2026-04-03
> >
> > Authors, thank you for your responses. I have no further questions at this time.

---

### Official Review · Reviewer_j6sf · 2026-03-13

**Soundness:** 2
**Presentation:** 2
**Significance:** 2
**Originality:** 2
**Overall Recommendation:** 4
**Confidence:** 3

**Summary:**

This paper proposes FLAC (Field Least-Energy Actor-Critic), a reinforcement learning algorithm for training iterative generative policies (e.g., flow/diffusion policies) under a maximum entropy objective. The key challenge here is that such policies usually lack tractable action log-densities, making entropy regularization difficult. The authors reinterpret maximum entropy RL through the Generalized Schrödinger Bridge (GSB) framework and show that the divergence from a reference stochastic process can be approximated by the kinetic energy of the policy velocity field. Based on this insight, FLAC introduces an energy-regularized actor objective and an automatic Lagrangian mechanism to control the energy budget.

**Compliance With Llm Reviewing Policy:**

Affirmed.

**Final Justification:**

The reply has addressed my problems.

**Key Questions For Authors:**

1. How strongly does the kinetic energy regularizer correlate with actual policy entropy during training?

2. In (6), why is the KL divergence irrelevant to Pref and P?

3. How to choose the reference stochastic process? Do different processes lead to different final losses?

**Limitations:**

yes

**Strengths And Weaknesses:**

Strength:
1. This paper provides an interesting connection between maximum entropy RL and the Schrödinger bridge.

2. FLAC avoids explicit likelihood or entropy estimation, which simplifies training for generative policies.

Weaknesses:

1. The algorithmic novelty is moderate; the method mainly replaces entropy regularization with an energy-based surrogate.

2. The paper lacks references to recent works in Iterative Generative Policies, such as [1, 2].

3. FLAC seems to be proposed based on the assumption that the maximum entropy distribution is a Gaussian distribution, since the mu_1 in paper should be a Gaussian.

[1] McAllister D, Ge S, Yi B, et al. Flow matching policy gradients[J]. arXiv preprint arXiv:2507.21053, 2025.

[2] Ding S, Hu K, Zhong S, et al. GenPO: Generative diffusion models meet on-policy reinforcement learning[J]. arXiv preprint arXiv:2505.18763, 2025.

---

> ### Author Rebuttal · Authors · 2026-03-30
>
> We sincerely thank the reviewer for the careful reading, and we appreciate the opportunity to clarify both the novelty and the other questions.
>
> > W1: On algorithmic novelty
>
> Thank you for raising this important concern. We would like to clarify the central idea of FLAC more directly.
>
> The main difficulty in applying MaxEnt RL to iterative generative policies is that the policy is defined implicitly through a transport process, making the action log-density intractable or complex. Existing approaches address this by introducing additional estimation machinery, such as variational objectives (DIME), auxiliary noise networks (SAC-FLOW), or multi-Gaussian estimators (DACER).
>
> FLAC takes **a fundamentally different route** by reformulating MaxEnt RL through a Generalized Schrödinger Bridge perspective. This perspective gives two key advantages:
>
> 1. **A new perspective under which the MaxEnt form can be recovered naturally.**  Under the condition stated in Proposition 4.1, the MaxEnt form can be recovered naturally from the bridge objective without requiring explicit entropy estimation.
> This offers a different angle from prior works, which focus on estimating entropy rather than examining what the underlying transport problem demands. Moreover, our derived optimal solution (Proposition 4.1) characterizes what property $\mu_1^{\mathrm{ref}}$ is needed to recover MaxEnt RL, this is a structural insight that was overlooked by prior methods.
> 2. **A simple algorithm.** Under this formulation, entropy regularization that previously required complex estimation reduces to penalizing kinetic energy, a quantity that can be simply given from the solver. The complexity is resolved at the formulation level, not by adding extra components.
>
> The GSB perspective makes MaxEnt RL directly accessible for implicit iterative generative policies. We were also encouraged that this perspective resonated with other comments as a novel or creative synthesis.
>
>
> > W2: Missing references
>
> Thank you for pointing this out. FPO and GenPO are relevant recent works that apply RL to iterative generative policies in the **on-policy** setting. FLAC addresses a complementary direction: **off-policy, value-based RL**. We will add these references and clarify this distinction in the revised version.
>
> > W3: On the Gaussian assumption
>
> Thank you for this question. We apologize if this point caused a misunderstanding. FLAC does **not** assume the maximum entropy policy is Gaussian. What Proposition 4.1 requires is that $\mu_1^{\mathrm{ref}}$ should be approximately uniform. Under this condition, the optimal terminal distribution recovers the form $\pi^*(a \mid s) \propto \mu_1^{\mathrm{ref}}(a)\exp(-G_s(a)/\alpha)$ (standard MaxEnt Boltzmann policy). We will make this distinction more explicit.
>
> > Q1: Kinetic energy and policy entropy correlation
>
> This is an excellent question. In fact, we can obtain the following **entropy lower bound**:
>
> $$H(\pi) \geq \log \mathrm{Vol}(\mathcal{A}) - \frac{\mathcal{E}_{\mathrm{KE}}}{2\sigma^2}$$
>
> Thus, constraining $\mathcal{E}_{\mathrm{KE}} \leq \bar{E}$ through the Lagrangian mechanism yields a corresponding **one-sided lower bound** on the terminal policy entropy. In this sense, FLAC does not require kinetic energy to exactly track entropy; rather, kinetic energy provides the key property needed here, namely an explicit guarantee against entropy collapse. We will add this bound in the revised paper.
>
> > Q2: In Eq. (6), why is the KL irrelevant to $P_{\mathrm{ref}}$ and $P$?
>
> We apologize if this point caused a misunderstanding: Eq. (6) is exactly the path-space KL between the controlled process $P$ and the reference process $P_{\mathrm{ref}}$:
>
> $$\mathrm{KL}(\pi \| \mu_1^{\mathrm{ref}}) \le \mathrm{KL}(P \| P_{\mathrm{ref}}) = \frac{1}{2\sigma^2}\\mathbb{E}\\left[\int_0^1 \|v_t\|^2\\mathrm{d}t\right]$$
>
> Here, $\sigma$ is the noise determined by $P_{\mathrm{ref}}$, while the kinetic-energy term depends on $P$ through its velocity $v_t$. **Thus the KL is relevant to $P_{\mathrm{ref}}$ and $P$**.
>
> > Q3: Choice of reference process and loss form
>
> Thank you for this question. In practice the choice is simple. Per Proposition 4.1, the essential requirement is that $\mu_1^{\mathrm{ref}}$ has approximately uniform distribution over the action domain. FLAC does not depend on a unique choice; any process meeting this condition yields an objective aligned with standard MaxEnt RL. For the loss form, whether in the SDE or the deterministic ODE regime, the resulting objective reduces to minimizing kinetic energy along the transport path.
>
> ---
>
> We thank the reviewer again and will address all points in the revision.

---

> > ### Author Rebuttal · Reviewer_j6sf · 2026-04-06
> >
> > Thanks for your reply. I will raise my score.

---

### Decision · Program_Chairs · 2026-04-30

**Decision:**

Accept (regular)

**Comment:**

This paper introduces Field Least-Energy Actor-Critic (FLAC), a framework designed to bring maximum entropy RL (MaxEnt RL) to iterative generative policies, such as diffusion and flow matching models. By reformulating policy optimization as a Generalized Schrödinger Bridge problem, the paper demonstrate that the entropy objective can be replaced by a physically grounded penalty on the kinetic energy of the velocity field. The result is a likelihood-free, "plug-and-play" algorithm that achieves competitive performance on high-dimensional locomotion tasks with significantly lower computational overhead than existing diffusion-based RL methods.

The reviewers appreciated the paper's theoretical elegance and its practical utility. Reviewer j6sf initially questioned the novelty of the approach, suggesting it might be a simple surrogate for entropy regularization, and raised concerns about the potential Gaussian assumptions underlying the method. However, the rebuttal clarified that the framework recovers the standard MaxEnt Boltzmann policy without Gaussian constraints, thus the reviewer upgraded their assessment.

The discussion phase has been helpful to provide additional technical details, particularly concerning the relationship between FLAC and existing state-of-the-art methods like DIME. Moreover, a rigorous ablation study was instrumental in convincing the reviewers of the method's robustness.